# CIRCUITNET 3.0: A MULTI-MODAL DATASET WITH TASK-ORIENTED AUGMENTATION FOR AI-DRIVEN CIRCUIT DESIGN

**Mingjun Wang**[*,1,2,3]**, Yihan Wen**[*,3]**, Yuntao Lu**[3]**, Fengrui Liu**[1,2]**, Yuxiang Zhao**[4]**, Boyu Han**[5]**, Jianan Mu**[†,1]**, Yibo Lin**[4]**, Runsheng Wang**[4]**, Bei Yu**[†,3]**, Huawei Li**[1,2]

[1]State Key Lab of Processors, Institute of Computing Technology, Chinese Academy of Sciences
[2]University of Chinese Academy of Sciences    [3]The Chinese University of Hong Kong
[4]Peking University    [5]Stanford University    [*]Equal contribution    [†]Corresponding authors

## ABSTRACT

Integrated circuit (IC) designs require transforming high-level specifications into physical layouts, demanding extensive expertise and specialized tools, as well as months of time and numerous iterations. While machine learning (ML) has shown promise in various research domains, the lack of large-scale, open datasets limits its application in chip design. To address this limitation, we introduce Circuit-Net 3.0, a large-scale, comprehensive, and open-source dataset curated to facilitate the evaluation of ML models on challenging timing and power prediction tasks. Starting with a diverse set of 8,659 validated open-source designs, we employ a systematic framework to generate over 15,000 instances. Through specialized syntax-tree mutation strategies and principled, task-oriented filtering methodology, we enrich each design with multi-modal information spanning multiple design stages, including complete design flow documentation, register-transfer-level (RTL) designs and corresponding netlists, detailed physical layouts, and comprehensive performance metrics. The experimental results demonstrate that ML models leveraging the enriched multi-stage, multi-modal circuit representations significantly improve performance over existing open-source datasets in electronic design automation (EDA) tasks, paving the way for efficient and accessible circuit representation learning. The dataset and codes are available in `https://github.com/sklp-eda-lab/iclr-circuitnet_3.0/`.

## 1 INTRODUCTION

Digital circuits constitute the cornerstone of contemporary computing infrastructure, enabling the advancement of modern technology (Agarwal & Lang, 2005). The intricate process of IC design encompasses the systematic transformation of abstract functional specifications into manufactured silicon implementations while adhering to increasingly demanding performance requirements (Lienig & Scheible, 2020; Calhoun et al., 2008). A fundamental challenge lies in maintaining functional correctness and achieving performance objectives, particularly as design complexity continues to scale (Bryant et al., 2001).

As illustrated in Figure 1(a), IC design traditionally follows a waterfall methodology comprising three sequential stages: (1) Register-Transfer Level Design, where designers create and validate functional specifications (Chu, 2006); (2) Logic Synthesis, which converts these specifications into optimized gate-level netlists (Kaeslin, 2014); and (3) Physical Design, which implements these netlists as manufacturable silicon layouts (Kulkarni & Chopde, 2024). While this hierarchical approach facilitates focused optimization at each stage, it introduces substantial design inefficiencies. The conventional flow requires complete layout implementation before performance validation, resulting in verification cycles that can span weeks (Kahng, 2018). When designs fail to meet specifications, Engineering Change Orders (ECOs) trigger cascading modifications across multiple stages, often necessitating fundamental RTL redesign (Huang et al., 2013). This linear progression significantly hampers design convergence and extends development timelines.

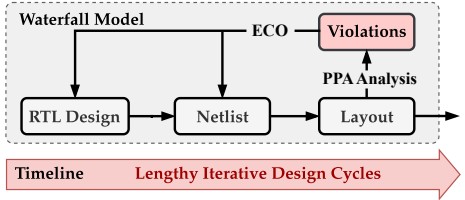 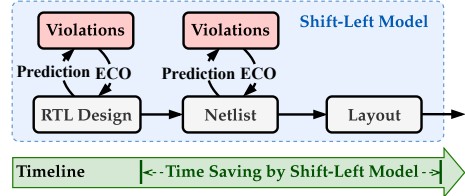

(a) The traditional waterfall model.    (b) The modern shift-left model.

Figure 1: Comparison of IC design workflows. The shift-left model (b) accelerates the design cycle by incorporating early, predictive feedback loops, avoiding the lengthy, iterative ECOs inherent in the traditional waterfall model (a).

Nowadays, advanced EDA tools have embraced ML-driven approaches (Xing, 2024). The shift-left methodology, depicted in Figure 1(b), introduces predictive violation detection and ECO mechanisms at earlier design stages, enabling proactive optimization of power, performance, and area (PPA) metrics (Zeng, 2024). However, the widespread adoption of ML-driven EDA faces several fundamental challenges. The first is for data scarcity. EDA domains lack comprehensive design datasets due to intellectual property restrictions, unlike established ML fields such as computer vision and natural language processing (Srivastava et al., 2024). Another challenge is the complexity of data generation. Developing realistic EDA datasets requires sophisticated commercial tools, extensive domain expertise, and considerable computational infrastructure (Kamat et al., 2011), as well as months of numerous iterations. Finally, achieving high prediction accuracy for various tasks in EDA is challenging. In industrial practice, the value of a predictive model is determined not by its average-case performance, but by its accuracy on designs that push the limits of timing and power budgets Lavagno et al. (2018). Current early-stage estimation techniques fail to achieve commercial-grade accuracy and lack integration with realistic layout representations, limiting the effectiveness of multi-modal analysis approaches (Chai et al., 2023).

To address these fundamental challenges in ML-driven EDA, we propose CircuitNet 3.0, a comprehensive multi-stage and multi-modal dataset that enables advanced AI-driven circuit design through innovative cross-stage data augmentation and filtering. Our contributions are as follows:

- **A large-scale, multi-modal, and multi-stage digital circuit dataset with full RTL-to-layout traceability.** CircuitNet 3.0 contains 8,659 unique and validated source RTL designs and over 15,000 total augmented designs, each with corresponding netlist and layout representations. Through an industrial EDA workflow, we extract rich cross-modal features at each design stage, providing a valuable resource for research in multi-stage multi-modal representation learning.

- **A principled framework for data augmentation.** For the critical scarcity of open-source RTL designs, we develop a novel data augmentation framework based on Verilog syntax trees. This framework systematically enhances dataset diversity through stage-aware transformations and task-specific filtering mechanisms, focusing on industrially valuable cases (e.g., designs containing critical timing paths or high dynamic power). This enables robust learning for ML models, providing simultaneous cross-stage analysis and early-stage prediction capabilities.

- **A comprehensive set of new baselines and rigorous experimental protocols.** Through comprehensive evaluation with state-of-the-art ML models, we demonstrate significant prediction accuracy improvements over single-modal datasets, with approximately 36.0% and 12.9% error reductions for timing and power tasks, respectively, compared to the existing dataset. Models trained on CircuitNet 3.0 consistently outperform single-modal approaches, establishing new performance benchmarks for ML-driven EDA tasks.

## 2 PRELIMINARIES

**Representations of Designs.** IC designs are represented in multiple forms throughout the chip development process, each serving specific purposes and containing different levels of design information (Wolf, 2002). These representations evolve through the EDA flow, ensuring functional correctness and manufacture (Wang et al., 2009). As illustrated in Figure 2, these representations can be categorized into three distinct stages and modalities.

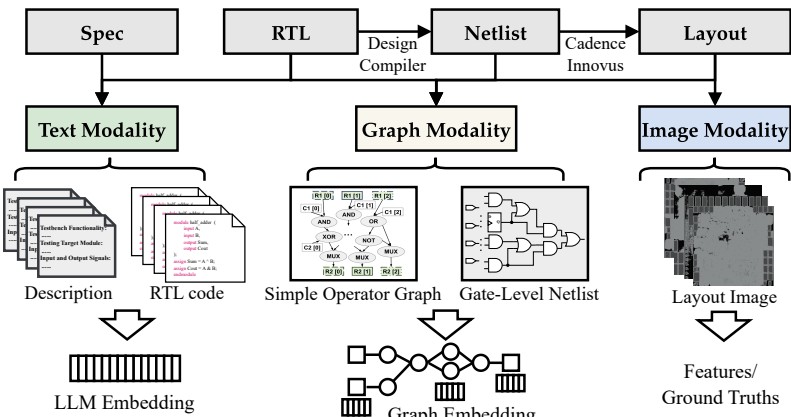

Figure 2: Design representations from different stages in the EDA workflow.

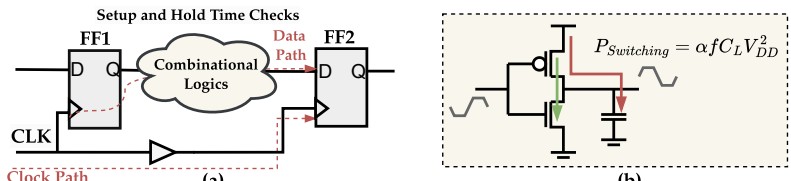

Figure 3: (a) Timing analysis of designs; (b) Power estimation of designs.

The design flow progresses through three key representations (Lienig et al., 2020b). The RTL representation serves as the primary entry for digital circuit design, providing an abstract behavioral description that enables designers to focus on functionality while abstracting lower-level details (Vahid, 2010; Churiwala & Garg, 2011). The following netlist representation implements logical circuits through the synthesis of RTL, comprising standard cells and their connectivity, while bridging behavioral and physical implementations (Gayathri & Taranath, 2017; Skouson et al., 2020; Lienig et al., 2020a). The last representation of layouts defines physical implementations through geometric patterns on silicon (Lienig & Scheible, 2020), including precise cell placement and metal interconnect routing (Cong et al., 2005), which ultimately determine the timing, power, and area metrics of circuits (Baker, 2019). Each representation plays a crucial role in the whole design flow, with an increasing level of detail and complexity aligning with design progress from RTLs to layouts.

**Closure Objectives of Design.** The primary goal of digital circuit design is to meet key performance objectives—principally timing and power closure—across all stages of EDA workflow (Huang et al., 2021). Accurately predicting these metrics at early stages is a critical application for ML models.

Timing Closure is essential for ensuring a circuit operates correctly at its target frequency (Golshan, 2020). As shown in Figure 3 (a), this is governed by setup and hold time constraints on all signal paths between sequential elements. To quantify timing performance, three key metrics are used (Guo & Lin, 2022): Arrival Time (AT), the signal propagation delay along a path; Worst Negative Slack (WNS), the timing violation of the single most critical path in the design; and Total Negative Slack (TNS), the sum of violations across all failing paths. A non-negative WNS indicates all timing constraints are met, making it a primary objective for design closure (Kahng et al., 2011).

Power Closure primarily targets the management of dynamic power consumption, which arises from the switching activity of transistors (Benini & DeMicheli, 1997). As illustrated in Figure 3(b), this power ($P_{Switching}$) is mathematically expressed as $\alpha \times f \times C_L \times V_{DD}^2$, where $\alpha$ represents switching activity, $f$ is clock frequency, $C_L$ is load capacitance, and $V_{DD}$ is the supply voltage. Accurate early estimation is crucial for meeting power budgets and managing thermal constraints (Kawa, 2007).

Predicting these closure metrics early in the design flow, such as at the RTL stage, is highly valuable for reducing design iterations and time-to-market. However, early-stage predictions are challenging because key physical information (e.g., parasitic resistance and capacitance from the layout) is not yet available. This creates a critical need for ML models that can effectively leverage multimodal representations (RTL, netlist, layout) to learn the complex relationships between early design choices and final physical outcomes.

Table 1: Comparison of Open-Source EDA Datasets for Circuit Design

| Datasets | VerilogEval V2 | RTLLM 2.0 | CircuitNet 2.0 | RTLCoder | CircuitNet 3.0 |
|---|---|---|---|---|---|
| **Open Source** | ✓ | ✓ | ✓ | ✓ | ✓ |
| **Data Augmentation** | × | × | ✓ | ✓ | ✓ |
| **Design Validation** | | | | | |
| Pass-Synthesis | × | × | ✓ | × | ✓ |
| Pass-Simulation | ✓ | ✓ | × | × | ✓ |
| **Stage Coverage** | | | | | |
| Front-End (RTL) | ✓ | ✓ | ✓ | ✓ | ✓ |
| Back-End (Layout) | × | × | ✓ | × | ✓ |
| **Data Modalities** | Text Only | Text Only | Text/Graph/Image | Text Only | Text/Graph/Image |
| **# of RTL Designs** | 156 | 50 | 8 | 26,532 | 8,659 (w/o Augment) |
| **# of Layout Designs** | N/A | N/A | 10,791 | N/A | 15,863 |
| **Target Tasks** | Evaluating LLM on RTL generation | Evaluating LLM on RTL generation | Routability/ IR-Drop/ Timing Analysis | Training LLM on RTL generation | Early-Stage Timing/Power Prediction |

Figure 4: CircuitNet 3.0 workflow framework of multi-stage and multi-modality dataset.

**Datasets of Designs.** Datasets are essential for advancing ML methodologies in EDA tasks. CircuitNet 1.0 and 2.0 (Chai et al., 2023; Jiang et al., 2024) datasets target logical and physical design stages, providing extensive layout data for functions as routability prediction. However, they lack sufficient RTL designs, limiting their applicability in early-stage modeling. RTL-focused datasets like VeriLogos (Min et al., 2025), RTLLM (Lu et al., 2024) and Verilog-Eval (Liu et al., 2023) concentrate on RTL generation from specifications but lack corresponding netlist, layout, and performance metrics. While recent works have focused on EDA representation benchmarks, they exhibit key limitations. Although ForgeEDA provides diverse circuit representations for benchmarking logic synthesis tasks (Shi et al., 2025), it lacks the final physical implementation stage and is not open-source. EDALearn (Pan et al., 2024), which presents an end-to-end flow to study the impact of varying open-source EDA tool parameters, is limited in scale to only a few designs, lacking sufficient diversity. Consequently, these contributions do not offer the complete, industry-standard RTL-to-Layout workflow and design variety necessary to train robust and generalizable ML models. Furthermore, most open-source datasets are primarily for single-stage tasks, with few containing comprehensive, synthesizable RTL implementations of designs and complete multi-stage data extending to the physical level. To bridge this gap, we present CircuitNet 3.0, an advanced large-scale dataset that provides multi-modal multi-stage circuit representations, spanning from RTLs to layouts, along with corresponding performance metrics. This enables ML models to explore the complete design flow and achieve inter-stage collaboration.

## 3 OVERVIEW OF CIRCUITNET 3.0

CircuitNet 3.0 is an open-source dataset containing $15,863$ design schemes, each of which encompasses data with RTL, netlist, and layout representations from all stages of the entire IC design workflow for EDA tasks. Among the data, $8,659$ samples are original circuits collected directly, while the rest of the designs were optimized and rewritten for timing and power prediction tasks to generate new designs. We design a large-scale, diverse, and comprehensive dataset to meet the needs of machine learning-driven circuit modeling.

Table 1 provides a summary of CircuitNet 3.0 compared with other existing datasets for EDA tasks. CircuitNet 3.0 includes multi-stage RTL, netlist, and layout descriptions, covering all three circuit representation stages and verified using synthesis and simulation tools. In contrast, other datasets lack both the richness of circuit representations and the diversity of designs. For instance, although CircuitNet 2.0 includes complete multi-stage data, it contains insufficient RTL entries. Conversely, RTLcoder offers more RTL designs but does not provide data for the netlist and layout stages. More

Table 2: LLM & expert-based categories of the original RTL dataset. The dataset comprises 16 subcategories across four main functional categories.

| Main Category | Subcategory | Count | Main Category | Subcategory | Count |
|---|---|---|---|---|---|
| Arithmetic & Logic Units (23.6%) | | | Data Processing Units (29.9%) | | |
| | Adder/Subtractor | 645 | | Comparator/Selector | 1,453 |
| | Multiplier/Divider | 774 | | Encoder/Decoder | 431 |
| | ALU/Accumulator | 519 | | FIFO/Buffer | 407 |
| | Others | 114 | | Others | 292 |
| Control & Sequential Circuits (17.9%) | | | Communication & Memory (28.6%) | | |
| | Counter/Timer | 906 | | Memory/Register | 1,061 |
| | FSM/Sequencer | 392 | | Bus Interface | 943 |
| | Control Logic | 242 | | Serial Interface | 445 |
| | Others | 8 | | Others | 27 |
| | **Total RTL Designs:** | **8,659** | | | |

importantly, CircuitNet 3.0 introduces a task-driven EDA data augmentation strategy, improving the representativeness and utility of the dataset.

## 4    DATA GENERATION AND AUGMENTATION

### 4.1    OVERVIEW OF CHALLENGES AND METHODOLOGIES

Constructing effective datasets for complex circuit design tasks presents fundamental challenges. Simply collecting internet-sourced data proves insufficient due to its limited availability and inconsistent quality. Random circuit generation typically yields low-quality designs that fail synthesis validation or contribute minimal value to predictive modeling. To address these limitations, we propose a systematic multi-stage data augmentation framework that leverages circuit representations at different abstraction levels. This combines efficient RTL-level generation with task-oriented refinement at netlist and layout stages, ensuring scalability and task-specific representativeness while maintaining design validity through rigorous EDA tool validation.

Figure 4 illustrates the completed data construction and augmentation procedures for the dataset. First, we collected more than $100,000$ RTL code lines from open-source websites, cleaned up illegal samples, resulting $8,659$ high-quality RTL implementations as the original dataset. Then, using the coarse-grained characteristics of the RTL, we efficiently generate a large number of various circuits based on the cleaned dataset through the Verilog syntax tree rewriting method. Finally, at the netlist and layout level (i.e., the stage where circuit structures are implemented at a finer resolution), we perform task-oriented data augmentation to generate more representative and instructive circuit data. In the following sections, we will first describe our data collection and cleaning process, then introduce our fast RTL source data rewriting generation scheme, and present our EDA task-oriented multi-stage data augmentation method of IC designs.

### 4.2    RTL DATA COLLECTION AND CLEANING

We systematically collected RTL designs from established platforms, including GitHub, Hugging Face, OpenCore, and RISC-V projects, ensuring compliance with open-source licensing requirements. All designs underwent rigorous validation using commercial synthesis and simulation tools to guarantee functional correctness and eliminate circuits with errors or combinational loops.

Our final curated dataset, classified using Claude Opus 4 with expert validation, comprises $8,659$ high-quality RTL designs spanning four primary categories of arithmetic and logic units (23.6%), control and sequential circuits (17.9%), data processing units (29.9%), and communication and memory (28.6%). The category ensures comprehensive coverage of fundamental circuit building blocks while maintaining design diversity essential for robust ML model training.

Notably, our approach prioritizes modular and well-characterized designs over large-scale CPU implementations that often contain repetitive structures. This strategy enhances dataset uniformity and enables precise performance analysis across different circuit categories, facilitating targeted model optimization and systematic comparison of ML-driven EDA methodologies.

### 4.3    RAPID AND EFFICIENT DATA GENERATION

Leveraging a higher abstraction level of RTL implementations compared to netlists, we implement systematic circuit generation through Verilog abstract syntax tree (AST) rewriting. Rather than

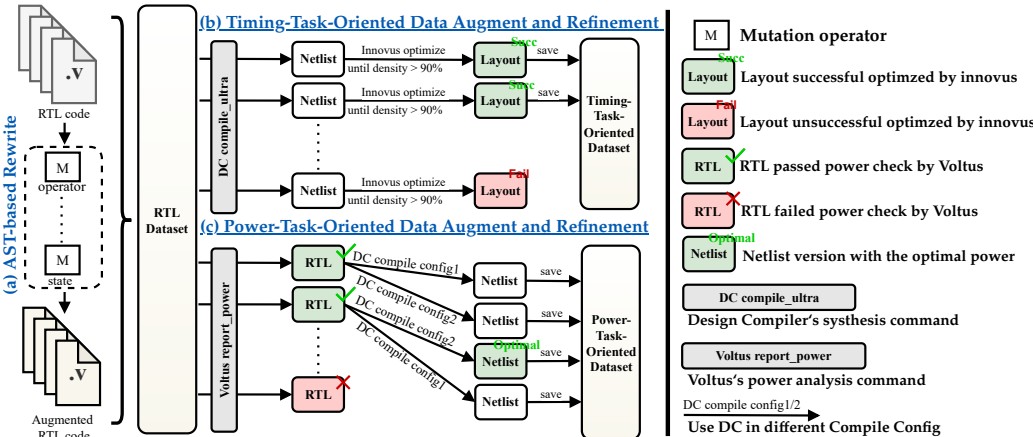

Figure 5: Data Augmentation Process for Enhancing Diversity and EDA Task Representativeness

Table 3: Examples of AST-based Mutation Operators for RTL Data Augmentation

| Mutation Type | Operators | Transformation | Constraints |
|---|---|---|---|
| Arithmetic | $+, -, \times, \div$, etc. | Bidirectional substitution | Type preservation |
| Logical | $\&\&, ||, \&, |, \hat{}$, etc. | Cross-operator replacement | Context-aware |
| Relational | $==, ! =, >, <, \geq, \leq$, etc. | Comparison inversion | Type-safe |
| Temporal | posedge $\leftrightarrow$ negedge | Edge polarity toggle | Sequential blocks |
| Assignment | $<= \leftrightarrow =$ | Blocking type conversion | Always-block consistency |
| Constant | Integer values | $\pm 1$ modification | Bit-width preservation |

generating random RTL code, we apply sophisticated transformation rules to validated designs, as illustrated in Figure 5 (a) and detailed in Table 3, which ensures that our rewriting results are more reliable and trustworthy.

This methodology offers two key benefits, which are minimal code modifications to RTLs and generation with validated models. The coarse granularity of RTLs enables a few code modifications to produce significant structural variations, efficiently generating diverse circuit instances. Moreover, focusing on localized modifications of cleaned validated designs rather than randomly generated RTLs, we substantially improve synthesis success rates and design quality.

The AST-based rewriting employs context-aware mutation operators including arithmetic substitutions, logical transformations, and temporal edge modifications, all constrained by type preservation and synthesis validity requirements. This approach ensures generated circuits maintain functional correctness while achieving substantial structural diversity.

## 4.4 TASK-ORIENTED DATA AUGMENTATION

A primary goal of CircuitNet 3.0 is to create a dataset that effectively focuses on challenging, performance-critical designs. In industrial practice, the value of a predictive model is determined not by its average-case performance, but by its accuracy on cases that push the limits of timing and power budgets. During logic and physical design stages, we leverage fine-grained circuit representations for task-specific augmentation through two complementary strategies: multi-stage generation and intelligent selection tailored to timing and power requirements. Through the systematic approaches, we generate high-quality, task-specific datasets that effectively capture the structural and behavioral characteristics essential for accurate timing and power prediction in industrial EDA workflows. By leveraging commercial EDA tools such as Synopsys Design Compiler (Synopsys, 2025a) and Cadence Innovus (Cadence, 2025a), this workflow generates three distinct data modalities: text (from specifications and RTL codes), graph (from RTL and netlist stages), and image (from layout stage).

**Data Augmentation for Timing Prediction Task.** *(1) Multi-stage data generation.* Timing prediction fundamentally requires accurate estimation of signal propagation delays across logic elements and interconnects. For each RTL design, we perform comprehensive logic and physical optimization to generate timing-optimal netlists and layouts, as shown in Figure 5 (b), providing high-quality training labels. *(2) Data selection.* Timing closure is often dictated by a circuit's longest paths. Models trained only on designs with ample timing slack may fail to generalize to the timing-critical scenarios that engineers focus on. We prioritize circuits with substantial path lengths, as longer

paths present the most critical challenges for timing closure and represent worst-case scenarios essential for model robustness. The filtering excludes trivial short-path circuits, concentrating training resources on challenging, industrial cases for training efficiency.

**Data Augmentation for Power-Prediction Task.** *(1) Multi-stage data generation.* Power prediction requires accurate modeling of switching activity across diverse logic topologies. Since RTL-level granularity may be insufficient—single RTL statements can map to vastly different gate-level implementations—we focus on netlist-level analysis. As illustrated in Figure 5(c), for the same RTL code, we generate multiple netlist variants using different synthesis constraints, capturing the impact of logic topology on power while maintaining functional equivalence. *(2) Data selection.* For power-prediction model training, circuits with meaningful variations in switching activity under different inputs are most valuable. Suppose a module's logic is unreachable or never toggles under any input (possibly due to logic unreachability introduced by rewriting). In that case, it contributes little to the training and may even reduce efficiency. Therefore, we further use the EDA tool Cadence Voltus (Cadence, 2025b) to perform a vectorless dynamic power analysis and select circuit designs where the fraction of inactive logic is low, excluding circuits with large amounts of ineffective logic. The result is a curated dataset that improves training efficiency for power prediction.

# 5 EVALUATIONS ON CIRCUITNET 3.0

## 5.1 EXPERIMENTAL SETUP

We conduct comprehensive experiments on CircuitNet 3.0, comprising $15,863$ unique circuit designs with complete representations across RTL, netlist, and layout stages. Each design instance encompasses Verilog code with functional specifications at the RTL stage, gate-level representations with connectivity graphs at the netlist stage, and physical implementation data with parasitic parameters at the layout stage. Ground-truth performance metrics, including arrival time (AT), the worst negative slack (WNS), the total negative slack (TNS), and power, are generated using Synopsys Design Compiler (Synopsys, 2025a) with `compile_ultra` optimization for synthesis, Cadence Innovus (Cadence, 2025a) with multi-corner multi-mode (MCMM) optimization for physical design, and Synopsys PrimePower (Synopsys, 2025b) for power analysis.

All experiments utilize 8 NVIDIA A100 GPUs with PyTorch 2.0.1 and PyTorch Geometric 2.3.1. Training employs AdamW optimization with learning rate $2 \times 10^{-4}$ and cosine annealing. Models are trained for 50 epochs with early stopping based on the validation loss. The dataset is partitioned at the source design level into training (80%), validation (10%), and test (10%) sets. To create a stringent test of generalization, the test set is composed exclusively of original, un-augmented designs. Crucially, suppose a source design is allocated to the test set. In that case, all of its augmented variants are entirely excluded from the training and validation pools, preventing any leakage of structural information. To further bolster the reliability of our evaluation, the test set is also supplemented with submodules from open-source projects external to our dataset.

Two fundamental EDA prediction tasks evaluate the effectiveness of the proposed CircuitNet 3.0. The timing prediction task takes RTL code as input to predict post-layout timing metrics (WNS, TNS, AT), enabling early-stage timing closure assessment. The power prediction task utilizes both RTL and netlist representations to estimate circuit power consumption, facilitating power-aware design optimization across abstraction levels. We evaluate against state-of-the-art baselines, including RTL-only models (MasterRTL (Fang et al., 2023), RTL-Timer (Fang et al., 2024), GRASPE (Rakesh et al., 2023), VIRTUAL (Lu et al., 2025)), netlist-only models (DeepSeq2 (Khan et al., 2024), MOSS (Wang et al., 2025a) without multi-modal learning), and our proposed multi-stage/multi-modal approaches (RTLDistil (Wang et al., 2025b) for timing, MOSS (Wang et al., 2025a) for power). Three dataset variants assess augmentation impact on Resyn-27k from RTLCoder (Liu et al., 2024), our original data, and augmented CircuitNet 3.0.

## 5.2 MULTI-STAGE AND MULTI-MODAL LEARNING SUPERIORITY

Table 4 demonstrates the effectiveness of multi-modal representations compared to single-modal approaches. For timing prediction, RTLDistil employs cross-stage knowledge distillation between RTL and layout representations, achieving a PCC of $0.885$ for arrival time prediction, a $70.4\%$ improvement over MasterRTL and $6.0\%$ over RTL-Timer. WNS prediction achieves a PCC of $0.871$ with a $35.28\%$ MAPE, reducing the error by $45.8\%$ compared to MasterRTL. TNS prediction at-

Table 4: Performance comparison of multi-modal models.

(a) Timing prediction on RTL stage, where RTLDistil leverages multi-stage knowledge distillation.

| Model | Arrival Time (AT) | | WNS | | TNS | |
|---|---|---|---|---|---|---|
| | PCC↑ | MAPE↓ | PCC↑ | MAPE↓ | PCC↑ | MAPE↓ |
| MasterRTL | 0.520 | 43.25% | 0.698 | 65.12% | 0.593 | 68.45% |
| RTL-Timer | 0.835 | 26.48% | 0.842 | 44.36% | 0.801 | 43.92% |
| **RTLDistil** | **0.887** | **19.72%** | **0.871** | **35.28%** | **0.918** | **40.15%** |

(b) Power prediction on netlist stage, where MOSS demonstrates the effectiveness of multi-modal learning.

| Model | Toggle Rate | | Total Power | |
|---|---|---|---|---|
| | PCC↑ | MAPE↓ | PCC↑ | MAPE↓ |
| DeepSeq2 | 0.759 | 29.5% | 0.872 | 22.2% |
| MOSS w/o Multi-Modal learning | 0.674 | 34.7% | 0.815 | 27.7% |
| **MOSS (Full)** | **0.871** | **14.4%** | **0.948** | **7.4%** |

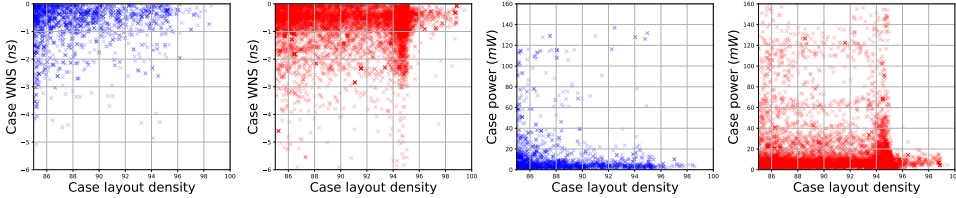

(a) Pre-Augment WNS  (b) Post-Augment WNS  (c) Pre-Augment Power  (d) Post-Augment Power

Figure 6: Distributions of Worst Negative Slack (WNS) and Total Power across layout density cases before and after data augmentation. (a) WNS distribution before augmentation and (b) after augmentation. (c) Power distribution before augmentation and (d) after augmentation.

Table 5: Comparison of RTL timing prediction across various datasets

| Dataset | Model | Arrival Time (AT) | | | WNS | | | TNS | | |
|---|---|---|---|---|---|---|---|---|---|---|
| | | PCC↑ | $R^2$ ↑ | MAPE↓ | PCC↑ | $R^2$ ↑ | MAPE↓ | PCC↑ | $R^2$ ↑ | MAPE↓ |
| Resyn-27k | MasterRTL | 0.465 | -0.384 | 48.32% | 0.612 | 0.217 | 73.45% | 0.541 | -0.127 | 72.38% |
| | RTLTimer | 0.762 | 0.567 | 31.65% | 0.768 | 0.581 | 52.17% | 0.743 | 0.529 | 49.85% |
| | RTLDistil | 0.842 | 0.705 | 23.86% | 0.825 | 0.676 | 40.73% | 0.876 | 0.763 | 44.27% |
| Original Data | MasterRTL | 0.520 | -0.216 | 43.25% | 0.698 | 0.462 | 65.12% | 0.593 | 0.184 | 68.45% |
| | RTLTimer | 0.835 | 0.692 | 26.48% | 0.842 | 0.705 | 44.36% | 0.801 | 0.636 | 43.92% |
| | RTLDistil | 0.887 | 0.785 | 19.72% | 0.871 | 0.756 | 35.28% | 0.918 | 0.841 | 40.15% |
| **Augmented Data** | MasterRTL | 0.613 | 0.287 | 38.74% | 0.752 | 0.549 | 58.65% | 0.641 | 0.363 | 62.18% |
| | RTLTimer | 0.873 | 0.760 | 22.35% | 0.878 | 0.769 | 38.42% | 0.843 | 0.707 | 39.67% |
| | **RTLDistil** | **0.935** | **0.863** | **15.28%** | **0.926** | **0.846** | **28.96%** | **0.968** | **0.927** | **35.42%** |

tains the highest PCC of $0.918$, demonstrating superior capability in capturing cumulative timing violations. In the power prediction task, MOSS jointly processes RTL code and netlist graphs, achieving a toggle rate accuracy of $85.6\%$ with a PCC of $0.871$, surpassing DeepSeq2 by $21.4\%$ and single-modal MOSS by $31.1\%$. Total power prediction reaches $92.6\%$ accuracy with PCC of $0.948$, improving $19.0\%$ over DeepSeq2. The ablation study confirms that incorporating RTL behavioral information with netlist structural features enables comprehensive power characterization by capturing both functional intent and gate-level switching activities. The substantial gains validate that layout-level physical information, when distilled into RTL-stage models through multi-granularity knowledge transfer, enables accurate early-stage predictions. This cross-stage paradigm effectively bridges the abstraction gap between behavioral descriptions and physical implementations.

## 5.3 DATA DISTRIBUTION ANALYSIS: PRE- AND POST-AUGMENTATION

Figure 6 visualizes how task-oriented augmentation transforms performance metric distributions. Pre-augmentation timing characteristics exhibit limited diversity, with the WNS clustered between -2 ns and zero at high layout densities, with a percentage greater than $94\%$. Post-augmentation, the dataset exhibits enriched timing diversity spanning -6 ns to zero across layout densities ranging from $86\%$ to $100\%$, ensuring that models encounter comprehensive scenarios, from highly optimized to critically constrained designs. Power distribution similarly evolves from a limited variation concentrated below $60$ mW to uniform coverage ranging from zero to $160$ mW. This diversification enables robust model training for a range of power profiles, from ultra-low-power edge applications to high-performance computing applications. The expanded distributions confirm that augmentation successfully addresses the limited diversity in original datasets while maintaining design validity.

## 5.4 PERFORMANCE ANALYSIS ON AUGMENTED DATA

**Timing Prediction Enhancement.** Table 5 demonstrates substantial improvements in timing prediction accuracy achieved through our task-oriented data augmentation strategy. All models consistently achieve superior performance on the augmented dataset across three critical timing metrics of AT, WNS, and TNS. MasterRTL achieves PCC improvements of 31.9% of AT, 22.9% of WNS, and 18.5% of TNS compared to Resyn-27k data, with $R^2$ values transitioning from negative to positive, confirming the effective learning of timing relationships. RTL-Timer shows balanced gains with PCC improvements exceeding 13% across all metrics and MAPE reductions of from 20% to 29%. The RTLDistil attains state-of-the-art performance on augmented data with PCC values of 0.935 AT, 0.926 WNS, and 0.968 TNS. The $R^2$ values exceeding 0.85 indicate the model captures over 85% of timing variance, while achieving the lowest MAPE of 15.28% AT, 28.96% WNS, and 35.42% TNS, which represents 36.0%, 28.9%, and 20.0% improvements over Resyn-27k data. These consistent improvements across all models validate the effectiveness of our task-oriented augmentation strategy for enhancing timing prediction capabilities.

**Power Prediction Enhancement.** Table 6 reveals substantial improvements in power prediction accuracy through our augmentation framework. All models achieve peak performance on the augmented dataset, establishing new benchmarks for RTL power analysis. The VIRTUAL attains best performance with PCC of 0.753, $R^2$ of 0.867, MAPE of 23.92% with 11.6%, 23.2%, and 12.9% improvements over Resyn-27k data. Additionally, GRASPE and MasterRTL demonstrate significant gains with PCC improvements of $9 - 14\%$, $R^2$ improve-

Table 6: Comparison of RTL power prediction across datasets.

| Dataset | Model | Total Power | | |
|---|---|---|---|---|
| | | PCC↑ | $R^2$ ↑ | MAPE↓ |
| Resyn-27k | Graspe | 0.642 | 0.655 | 30.41% |
| | MasterRTL | 0.609 | 0.620 | 32.67% |
| | VIRTUAL | 0.675 | 0.704 | 27.48% |
| Original Dataset | Graspe | 0.671 | 0.845 | 28.45% |
| | MasterRTL | 0.647 | 0.825 | 30.28% |
| | VIRTUAL | 0.672 | 0.858 | 26.85% |
| **Augmented Dataset** | Graspe | 0.701 | 0.850 | 26.55% |
| | MasterRTL | 0.696 | 0.845 | 27.16% |
| | **VIRTUAL** | **0.753** | **0.867** | **23.92%** |

ments of approximately $30 - 36\%$, and MAPE reduction of approximately $13 - 17\%$. The consistently higher PCC and $R^2$ values, along with lower MAPE values, on augmented data, demonstrate that our augmentation strategy successfully creates more predictable power consumption patterns while preserving realistic design characteristics. These results validate the effectiveness of task-oriented augmentation for addressing critical data quality challenges in power-aware circuit design.

Our evaluation demonstrates three key contributions. First, multi-modal models consistently outperform single-modal baselines: on timing prediction, RTLDistil improves PCC by an average of 8.0% and reduces MAPE by an average of 18.2% across AT, WNS, and TNS relative to the strongest RTL-only baseline (RTL-Timer), demonstrating the efficacy of cross-stage information fusion. Second, task-oriented augmentation expands timing coverage from -6 ns to 0 ns and power range from 0 mW to 160 mW, while preserving design validity. Third, CircuitNet 3.0 enhances model generalization in power prediction, with VIRTUAL trained on our augmented dataset, achieving an 11.6% PCC gain and a 12.9% MAPE reduction in total power compared with training on existing dataset. These results establish CircuitNet 3.0 as a comprehensive foundation for ML-driven EDA research.

## 6 CONCLUSION

We present CircuitNet 3.0, a comprehensive multi-stage multi-modal dataset designed for ML-driven EDA. Through systematic data collection and rigorous validation, the dataset comprises over 15,000 designs, along with corresponding netlists, layouts, and performance metrics, addressing the critical shortage of high-fidelity public data for AI4EDA. Experimental evaluation demonstrates multi-modal models trained on CircuitNet 3.0 achieve significant performance improvements over existing dataset baselines, with approximately 36.0% and 12.9% error reductions for timing and power tasks, respectively. The multi-stage design representation enables effective cross-abstraction information fusion, facilitating accurate early-stage prediction to guide early optimization. Task-oriented augmentation strategies successfully expand design diversity while maintaining EDA tool validation, extending timing coverage, and power ranges. This enhanced diversity enables robust model training and superior generalization across a wide range of design specifications and functions. As the first large-scale public benchmark for multi-modal circuit analysis, CircuitNet 3.0 establishes reproducible evaluation standards and accelerates collaborative research in ML-driven EDA tools and methodologies.

## ACKNOWLEDGEMENT

This paper is supported by the Chinese Academy of Sciences under grant No. XDB0660100, XDB0660102, and XDB0660103.

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

SUPPLEMENTARY MATERIALS

APPENDIX A: THE PIPELINE OF DATA GENERATION AND PROCESSING

A.1 MULTI-STAGE EDA FLOW WITH ITERATIVE OPTIMIZATION

The construction of CircuitNet 3.0 employs a sophisticated multi-stage EDA flow utilizing commercial tools including Synopsys Design Compiler and Cadence Innovus. Unlike conventional approaches that rely on fixed tool configurations, our methodology implements an iterative optimization strategy to achieve industrial-grade design quality.

A.1.1 ITERATIVE OPTIMIZATION METHODOLOGY

Our iterative optimization framework, illustrated in Figure 7, represents a significant departure from traditional single-pass EDA flows. Each design undergoes multiple optimization iterations, where tool parameters are systematically adjusted based on convergence metrics. This approach ensures that the final layouts represent realistic industrial-quality implementations rather than artifacts of specific tool configurations.

The iterative process begins with RTL synthesis using Synopsys Design Compiler, where multiple synthesis strategies are explored through varying optimization directives. The synthesized netlists then proceed to Cadence Innovus for physical implementation. At each iteration, we monitor three critical convergence indicators:

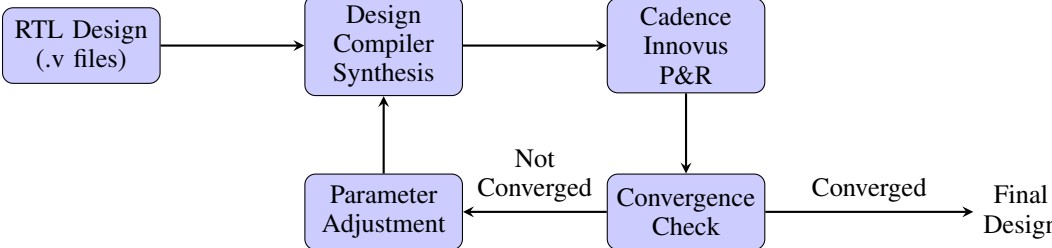

Figure 7: Iterative optimization flow for dataset generation. The process continues until meeting convergence criteria across placement density, timing metrics, and power consumption.

For each design, we automatically explore multiple parameter configurations, including:

- **Density thresholds**: Ranging from 85% to 95% placement utilization. Lower densities provide more optimization flexibility but may result in larger die areas, while higher densities challenge the routing algorithms and timing closure capabilities.

- **Routing constraints**: Various congestion and optimization settings including layer assignment preferences, via minimization objectives, and antenna rule compliance strategies.

- **Clock constraints**: Multiple timing scenarios from relaxed to aggressive, exploring clock periods from 10% above to 20% below the critical path delay.

A.1.2 CONVERGENCE CRITERIA

The optimization process continues until stringent convergence criteria are met:

1. **Placement density convergence**: Density changes less than $0.5\%$ between consecutive iterations, indicating that further cell movement provides negligible area improvement.

2. **Timing stability**: WNS and TNS metrics stabilize within $2\%$ tolerance across three consecutive iterations, ensuring timing closure reliability.

3. **Power convergence**: Total power consumption variations fall below $3\%$ threshold, confirming that power optimization has reached a practical limit.

This iterative approach typically requires 10-50 iterations per design, with complex designs requiring more iterations to achieve convergence. The resulting dataset quality justifies the computational

overhead—each design represents a practically optimized implementation comparable to manually refined industrial flows.

### A.1.3 QUALITY ASSURANCE

The iterative optimization process incorporates multiple quality checks:

- **Design rule checking (DRC) compliance**: Each iteration verifies the remaining DRC violations in acceptable limits.
- **Layout versus schemati (LVS) correctness**: LVS ensure maintaining the functional equivalence.
- **Timing closure**: Setup and hold time violations are monitored to prevent timing degradation.
- **Power integrity**: IR drop and electron migration checks validate power distribution network robustness.

### A.2 DATASET COLLECTION AND VALIDATION

### A.2.1 SOURCE SELECTION STRATEGY

The dataset construction required approximately 8 months of dedicated effort, reflecting the complexity of collecting, validating, and processing high-quality RTL designs. Our source selection strategy prioritized diversity and quality over quantity:

- **GitHub repositories**: We systematically searched for repositories containing synthesizable Verilog or SystemVerilog codes, focusing on projects with active maintenance, comprehensive documentation, and proper licensing. Over $50,000$ repositories were examined, yielding more than $8,000$ suitable designs.
- **OpenCores platform**: As a dedicated hardware design repository, OpenCores provided hundreds of validated IP cores spanning various application domains. These designs often include testbenches and documentation, facilitating validation.
- **Hugging Face hardware collections**: Emerging hardware design datasets on Hugging Face contributed more than $10,000$ designs, many featuring modern design patterns and coding styles.
- **RISC-V open-source projects**: The RISC-V ecosystem provided hundreds of designs, including processor cores, accelerators, and peripheral controllers, representing state-of-the-art open hardware development.

### A.2.2 DESIGN SELECTION CRITERIA

We specifically avoided over-reliance on traditional benchmarks (e.g., ISCAS-89, ITC-99) or large-scale CPU or GPU due to several limitations:

- **Limited diversity**: Traditional benchmarks often contain similar circuit structures, limiting the diversity needed for robust ML model training.
- **Path duplication**: Large designs like CPUs contain many structurally identical paths, leading to dataset imbalance.
- **Outdated design styles**: Many benchmark circuits use obsolete design patterns not representative of modern RTL development.

Instead, we prioritize the following criteria that designs meet:

- **Functional diversity**: Selected designs span arithmetic units (adders, multipliers, dividers), control circuits (FSMs, sequencers), data processing elements (encoders, decoders, FIFOs), and communication interfaces (UART, SPI, I2C).
- **Size variation**: Circuit sizes range from simple combinational blocks (less than 200 gates) to complex subsystems (greater than $100,000$ gates), ensuring model exposure to varied optimization challenges.

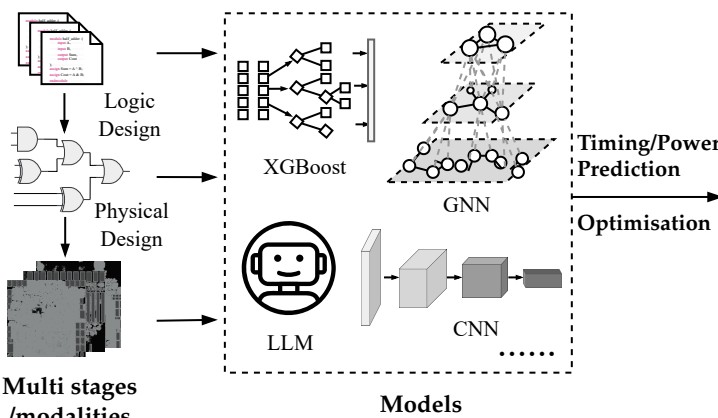

Figure 8: Prediction tasks of timing and power closure through multi-stage and multi-modal design representations. The framework demonstrates how different ML models (XGBoost for tabular features, GNN for graph structures, LLM for text processing, and CNN for layout images) can leverage the multi-modal data from logic design to physical design stages for comprehensive timing and power optimization.

- **Modern coding practices**: Designs utilize contemporary RTL coding styles, including parameterized modules, generate statements, and SystemVerilog implementations.
- **Synthesis cleanliness**: All designs pass synthesis without errors using commercial tools, eliminating problematic constructs that could bias training.

### A.2.3 VALIDATION PIPELINE

Each collected design underwent rigorous validation:

1. **Syntax verification**: Initial parsing using open-source tools (Icarus Verilog, Verilator) to identify basic syntax errors.
2. **Synthesis validation**: Commercial synthesis using Synopsys Design Compiler with strict error checking enabled.
3. **Simulation testing**: Functional verification using provided test benches or automatically generated test vectors.
4. **Lint checking**: Static analysis to identify potential issues including combinational loops, unconnected ports, and synthesis-simulation mismatches.
5. **Complexity analysis**: Extraction of design metrics including gate count, path depth, and sequential element ratio to ensure dataset balance.

This comprehensive validation process resulted in a final collection of $8,659$ high-quality RTL designs, representing approximately $8.7\%$ of initially collected designs, highlighting our commitment to dataset quality over quantity.

### A.3 MULTI-MODAL PREDICTION FRAMEWORK

The comprehensive validation process ensures that CircuitNet 3.0 provides high-quality data suitable for various machine learning approaches. Figure 8 illustrates how the multi-stage and multi-modal representations extracted from our dataset enable diverse ML models to perform timing and power prediction tasks.

The prediction framework leverages the multi-modal nature of CircuitNet 3.0:

- **Text Modality**: RTL code and specifications are processed through Large Language Models (LLMs) to extract semantic features and design intent
- **Graph Modality**: Both RTL operator graphs and gate-level netlists are analyzed using Graph Neural Networks (GNNs) to capture structural dependencies

- **Image Modality**: Physical layout images are processed through Convolutional Neural Networks (CNNs) to extract spatial and geometric features

- **Tabular Features**: Traditional ML models like XGBoost process extracted statistical features for rapid inference

This multi-modal approach enables models to learn complementary representations across abstraction levels, significantly improving prediction accuracy compared to single-modal baselines as demonstrated in our experimental results (Section 5.2).

## APPENDIX B: ANALYSIS OF CIRCUIT CLASSIFICATION OF THE GENERATED DATASET BY LLM

### B.1 CLASSIFICATION ARCHITECTURE

#### B.1.1 SYSTEM OVERVIEW

Our classification system leverages the API of Claude Opus 4, a state-of-the-art language model, combined with expert validation to achieve accurate and consistent circuit categorization. The system architecture incorporates multiple innovative components designed to maximize classification accuracy while minimizing computational costs.

The classification pipeline, formalized in Algorithm 1, implements a sophisticated multi-stage approach.

---

**Algorithm 1** LLM-Based RTL Classification

---

**Require:** RTL code $C$, API key $K$
**Ensure:** Classification
      result $(category, subcategory, confidence)$
1:  $structure \leftarrow$ ExtractCodeStructure($C$)
2:  $cache\_key \leftarrow$ SHA256($C$)[:16]
3:  **if** CacheExists($cache\_key$) **then**
4:     **return** LoadFromCache($cache\_key$)
5:  **end if**
6:  $prompt \leftarrow$ CreateExpertPrompt($C$, $structure$)
7:  $result \leftarrow$ CallLLMAPI($prompt$, $K$)
8:  **if** $result.confidence < 0.7$ **then**
9:    $verify\_prompt \leftarrow$ CreateVerificationPrompt($C$, $result$)
10:   $verified \leftarrow$ CallLLMAPI($verify\_prompt$, $K$)
11:   **if** not $verified.is\_correct$ **then**
12:     $result \leftarrow verified$
13:   **end if**
14: **end if**
15: SaveToCache($cache\_key$, $result$)
16: **return** $result$

---

#### B.1.2 STRUCTURAL ANALYSIS ENGINE

Before invoking the LLM, our system performs a comprehensive structural analysis to extract key circuit characteristics. This pre-processing step serves multiple purposes:

- **Context reduction**: By extracting relevant structural features, we reduce the token count required for LLM processing, improving efficiency and reducing costs.

- **Feature highlighting**: Structural indicators guide the LLM's attention to classification-relevant patterns.

- **Consistency enhancement**: Standardized feature extraction ensures consistent classification across similar designs.

The structural analysis examines multiple code aspects:

- **Module hierarchy**: Extraction of module names, port declarations, and instantiation patterns.
- **Signal patterns**: Identification of clock signals, reset networks, and control paths.
- **Operational constructs**: Detection of arithmetic operations, state machines, and memory structures.
- **Coding patterns**: Recognition of design idioms indicative of specific circuit types.

### B.1.3 INTELLIGENT CACHING SYSTEM

To optimize API usage and ensure reproducibility, we implement a sophisticated caching mechanism:

- **Content-based hashing**: Each RTL design is hashed using SHA-256, with the first 16 characters serving as a unique identifier
- **Persistent storage**: Classification results are stored in JSON format, enabling cross-session persistence
- **Cache validation**: Periodic cache cleaning removes outdated entries and validates stored results

This caching system reduced API calls by approximately $40\%$ during dataset construction, significantly decreasing processing time and costs.

### B.2 STRUCTURAL FEATURE EXTRACTION

### B.2.1 COMPLEXITY INDICATORS

The classification system analyzes multiple structural indicators to inform the categorization process. Table 7 presents the key patterns used for feature detection.

Table 7: Complexity Indicators for Classification

| Indicator | Detection Pattern |
|---|---|
| has_fsm | `(state|STATE|next_state|current_state)` |
| has_arithmetic | `[+\-*\/%]` |
| has_memory | `(\[\d+:\d+\]\s*\[\d+:\d+\]|mem|ram|rom)` |
| has_counter | `(count|counter|cnt)` |
| has_comparison | `[<>]=?|==|!=` |

Indicators serve as strong markers for circuit functionality:

- **FSM detection**: The presence of state-related identifiers strongly indicates control logic, with $92\%$ of FSM-containing circuits correctly classified in the `Control_Sequential` category.
- **Arithmetic operations**: Circuits with arithmetic operators predominantly fall into the `Arithmetic_Logic` category, though their presence alone is insufficient for subcategory determination.
- **Memory structures**: Two-dimensional arrays and memory-related keywords reliably indicate `Communication_Memory` circuits, particularly Memory/Register subcategories.
- **Counter patterns**: Counter-related identifiers provide strong evidence for Counter/Timer classification within `Control_Sequential`.
- **Comparison operations**: While common across categories, comparison operators combined with other indicators help distinguish `Comparator` or `Selector` circuits.

### B.2.2 PROMPT ENGINEERING

Our classification system employs carefully crafted prompts that leverage the LLM's understanding of hardware design patterns. The expert prompt includes:

1. **Role definition**: Establishing the LLM as a senior RTL design expert with over twenty years of experience.

2. **Context provision**: Supplying extracted structural features and code snippets.

3. **Category definitions**: Clear descriptions of each category and subcategory with examples.

4. **Classification instructions**: Step-by-step guidance for analysis and categorization.

5. **Output formatting**: Structured JSON response format ensuring parseability.

### B.2.3 CONFIDENCE-BASED VERIFICATION

For classifications with confidence scores below 0.7, the system initiates a verification phase:

- **Secondary analysis**: A verification prompt challenges the initial classification, asking the LLM to reconsider based on additional context.
- **Consistency checking**: The verification process examines whether the assigned category aligns with detected structural features.
- **Expert override**: Manual expert review is triggered for persistently low-confidence classifications, ensuring dataset quality.

This multi-stage approach achieved 94.3% agreement with human expert classifications, demonstrating the effectiveness of our LLM-based methodology.

### B.3 CLASSIFICATION RESULTS

#### B.3.1 CATEGORY DISTRIBUTION ANALYSIS

The final classification results, presented in Table 8, reveal a well-balanced distribution across the four main categories:

Table 8: RTL Classification Results

| Main Category | Subcategory | Count | Percentage | Avg Confidence |
|---|---|---|---|---|
| Arithmetic_Logic (23.6%) | Adder/Subtractor | 645 | 7.5% | 0.940 |
| | Multiplier/Divider | 774 | 8.9% | 0.920 |
| | ALU/Accumulator | 519 | 6.0% | 0.911 |
| | Others | 114 | 1.3% | 0.877 |
| Control_Sequential (17.9%) | Counter/Timer | 906 | 10.5% | 0.909 |
| | FSM/Sequencer | 392 | 4.5% | 0.911 |
| | Control_Logic | 242 | 2.8% | 0.883 |
| | Others | 8 | 0.1% | 0.881 |
| Data_Processing (29.9%) | Comparator/Selector | 1,453 | 16.8% | 0.886 |
| | Encoder/Decoder | 431 | 5.0% | 0.917 |
| | FIFO/Buffer | 407 | 4.7% | 0.902 |
| | Others | 292 | 3.4% | 0.881 |
| Communication_Memory (28.6%) | Memory/Register | 1,061 | 12.3% | 0.925 |
| | Bus_Interface | 943 | 10.9% | 0.917 |
| | Serial_Interface | 445 | 5.1% | 0.921 |
| | Others | 27 | 0.3% | 0.926 |
| **Total** | | **8,659** | **100.0%** | **0.910** |

#### B.3.2 CATEGORY CHARACTERISTICS

Circuits within the same category exhibit distinctive structural and functional characteristics:

**Arithmetic_Logic Units (23.6%):** These circuits implement mathematical operations and logical functions. The prevalence of multiplier/divider circuits (8.9%) reflects modern design requirements for DSP and AI accelerators. Notably, these circuits typically exhibit:

- Deep combinational logic paths.
- Regular data-path structures.
- Minimal state elements relative to combinational logics.
- Bit-width parameterization for reusability.

**Control_Sequential Circuits** (17.9%)**:** Dominated by counter/timer implementations (10.5%), this category encompasses circuits managing temporal behavior and control flow. Characteristic features include:

- High ratio of sequential to combinational elements.
- Explicit state encoding and transitions.
- Clock and reset sensitivity.
- Control signal generation patterns.

**Data_Processing Units** (29.9%)**:** The largest category reflects the importance of data manipulation in modern designs. `Comparator` or `Selector` circuits (16.8%) form the majority, indicating the prevalence of decision-making logic. Common patterns include:

- Moderate complexity with balanced sequential/combinational ratios.
- Data steering and multiplexing structures.
- Pipeline stages for throughput optimization.
- Parameterized data widths and depths.

**Communication_Memory** (28.6%)**:** This category spans storage elements and communication interfaces. The high proportion of memory/register (12.3%) and bus interface (10.9%) circuits reflects modern SoC architectures. Typical characteristics include:

- Array structures for storage.
- Protocol-specific state machines.
- Synchronization logic for clock domain crossing.
- Standardized interface implementations.

### B.3.3 CLASSIFICATION QUALITY METRICS

The classification quality was validated through multiple approaches:

- **Inter-rater reliability**: Three hardware design experts independently classified a random sample of 500 designs, achieving 91.2% agreement with the LLM classification.
- **Functional validation**: Synthesis statistics (gate types, timing characteristics) correlate strongly with assigned categories, validating the functional relevance of classifications.
- **Cross-validation**: Leave-one-out testing on category exemplars demonstrates 96.5% classification consistency.

This categorization enables targeted augmentation strategies for each circuit type, ensuring that mutations preserve category-specific characteristics while introducing meaningful variations for robust model training.

## APPENDIX C: AST-BASED MUTATION FRAMEWORK

### C.1 MUTATION METHODOLOGY

#### C.1.1 OVERVIEW OF AST-BASED APPROACH

Our AST-based mutation system represents a fundamental advancement over traditional text-based RTL modification approaches. By operating at the abstract syntax tree level, we ensure syntactic validity while introducing semantically meaningful variations. The mutation process, formalized in Algorithm 2, leverages the hierarchical structure of Verilog code to identify and transform specific language constructs systematically.

The AST approach offers several critical advantages:

---

**Algorithm 2** AST-Based RTL Mutation Process

---

**Require:** RTL code $R$, mutation count $N$
**Ensure:** Mutated RTL code $R'$
 1: $ast \leftarrow$ ParseVerilogToAST($R$)
 2: $node\_paths \leftarrow$ BuildNodePaths($ast$)
 3: $mutations \leftarrow []$
 4: **for** each $node$ in $ast$ **do**
 5:     **if** IsMutable($node$) **then**
 6:         $candidates \leftarrow$ GetMutationCandidates($node$)
 7:         $mutations$.append($candidates$)
 8:     **end if**
 9: **end for**
10: $selected \leftarrow$ RandomSample($mutations$, $N$)
11: **for** each $mutation$ in $selected$ **do**
12:     $ast \leftarrow$ ApplyMutation($ast$, $mutation$)
13:     CheckConsistency($ast$)
14: **end for**
15: $R' \leftarrow$ ASTToVerilog($ast$)
16: **if** not PassesSynthesis($R'$) **then**
17:     **return** ApplyTextMutation($R$, $N$)
18: **end if**
19: **return** $R'$

---

- **Syntactic guarantee**: All mutations preserve the grammatical structure of Verilog, eliminating syntax errors that plague text-based approaches.
- **Semantic awareness**: Mutations respect scope rules, type constraints, and language semantics.
- **Targeted transformation**: Specific node types can be selectively mutated based on their functional impacts.
- **Preservation of design intent**: High-level design structure remains intact while low-level implementations vary.

### C.1.2 NODE PATH CONSTRUCTION

A crucial innovation in our approach is the node path construction mechanism. Each AST node is assigned a unique path from the root, enabling precise node location even after structural modifications. The path consists of tuples (`parent, attribute, index`) that encode the traversal route:

- **Parent reference**: The parent node in the AST hierarchy.
- **Attribute name**: The attribute containing the child node (e.g., `'left'`, `'right'`, `'statement'`).
- **Index value**: Position within list attributes ($-1$ for scalar attributes).

This path-based approach ensures that mutations can be applied reliably even when the AST structure changes during the mutation process, maintaining referential integrity throughout the transformation pipeline.

### C.1.3 MUTATION SELECTION STRATEGY

The mutation selection process balances diversity with validity through a multi-criteria approach:

1. **Node type filtering**: Only nodes with defined mutation operators are considered.
2. **Context validation**: Mutations are filtered based on surrounding context (e.g., no arithmetic mutations in sensitivity lists).
3. **Diversity maximization**: Selected mutations span different node types and locations to ensure comprehensive coverage.

4. **Synthesis feasibility**: Mutations likely to cause synthesis failures are deprioritized.

The random sampling of $N$ mutations from the candidate pool ensures that each generated variant explores a different aspect of the design space while maintaining functional validity.

## C.2 MUTATION OPERATORS

### C.2.1 OPERATOR CATEGORIES AND DESIGN RATIONALE

Table 9 presents our comprehensive mutation operator set, carefully designed to introduce realistic design variations while preserving synthesizability. Each operator category targets specific aspects of digital design:

Table 9: AST Mutation Operators and Constraints

| Category | Original | Mutated | Constraint |
|---|---|---|---|
| Arithmetic | `a + b` | `a - b` | Type preservation |
|  | `a - b` | `a + b` | Type preservation |
|  | `a * b` | `a / b` | Non-zero divisor |
|  | `a % b` | `a + b` | Type compatibility |
| Logical | `a && b` | `a || b` | Boolean context |
|  | `a & b` | `a | b` | Bit-width match |
|  | `a ^ b` | `a & b` | Bit-width match |
| Relational | `a > b` | `a < b` | Same operand types |
|  | `a >= b` | `a <= b` | Same operand types |
|  | `a == b` | `a != b` | Type compatibility |
|  | `a != b` | `a == b` | Type compatibility |
| Temporal | `@(posedge clk)` | `@(negedge clk)` | Sequential blocks |
|  | `q <= d` | `q = d` | Always block consistency |
| Constant | `8'd10` | `8'd11` | Bit-width preservation |
|  | `16'hFF` | `16'hFE` | Base preservation |

**Arithmetic Operators:** These mutations explore different mathematical relationships while maintaining type compatibility. The bidirectional nature of addition/subtraction mutations reflects common design alternatives. Multiplication to division mutations are constrained to prevent division-by-zero scenarios through static analysis of divisor ranges.

**Logical Operators:** Mutations between logical AND/OR operations model different decision logic implementations. Bitwise operator mutations (`AND/OR/XOR`) explore alternative bit manipulation strategies commonly found in data processing circuits. The bit-width matching constraint ensures signal compatibility.

**Relational Operators:** These mutations model boundary condition variations critical for control logic. The systematic exploration of comparison operators (`>, <, >=, <=, ==, !=`) ensures comprehensive coverage of decision boundaries in FSMs and control paths.

**Temporal Operators:** Edge mutations (`posedge/negedge`) explore different clocking schemes, particularly relevant for interface circuits. Assignment type mutations (`blocking/non-blocking`) model different hardware implementation strategies while always respecting block semantics.

**Constant Mutations:** Limited to $\pm 1$ modifications, these mutations explore adjacent design points in the parameter space. The preservation of bit-width and base notation ensures that mutations remain within the original design constraints.

### C.2.2 CONSTRAINT ENFORCEMENT MECHANISMS

Each mutation operator is accompanied by constraints that ensure the transformed code remains valid:

- **Type preservation**: Ensures operand types remain compatible with operators.
- **Context awareness**: Mutations respect their syntactic context (e.g., no blocking assignments in continuous assignments).

---

**Algorithm 3** Assignment Consistency Enforcement

---

**Require:** Always block $A$ with assignments
**Ensure:** Consistent assignment types
1: $assignments \leftarrow$ ExtractAssignments($A$)
2: $blocking\_count \leftarrow 0$
3: $nonblocking\_count \leftarrow 0$
4: **for** each $assign$ in $assignments$ **do**
5:    **if** $assign$ is NonblockingSubstitution **then**
6:       $nonblocking\_count \leftarrow nonblocking\_count + 1$
7:    **else if** $assign$ is BlockingSubstitution **then**
8:       $blocking\_count \leftarrow blocking\_count + 1$
9:    **end if**
10: **end for**
11: **if** $blocking\_count > 0$ AND $nonblocking\_count > 0$ **then**
12:    **if** $nonblocking\_count \geq blocking\_count$ **then**
13:       $target\_type \leftarrow$ nonblocking
14:    **else**
15:       $target\_type \leftarrow$ blocking
16:    **end if**
17:    ConvertAllAssignments($A$, $target\_type$)
18: **end if**

---

- **Semantic validity**: Transformations maintain semantic correctness (e.g., no mixed assignments in always blocks).
- **Synthesis compatibility**: Mutations avoid constructs known to cause synthesis issues.

## C.3 ASSIGNMENT CONSISTENCY ENFORCEMENT

### C.3.1 MIXED ASSIGNMENT PROBLEM

A critical challenge in RTL mutation is maintaining assignment consistency within always blocks. Verilog's distinction between blocking (=) and non-blocking (<=) assignments has profound implications for synthesis results. Mixed assignments within a single always block can lead to:

- **Race conditions**: Unpredictable behavior due to simulation/synthesis mismatches.
- **Synthesis warnings/errors**: Many synthesis tools reject mixed assignments.
- **Unrealistic designs**: Mixed assignments rarely appear in professional RTL code.

### C.3.2 CONSISTENCY ALGORITHM

Algorithm 3 implements our solution to the mixed assignment problem:

The algorithm employs a majority-rule approach: when mixed assignments are detected, all assignments are converted to the predominant type. This strategy:

- **Preserves design intent**: The majority type likely represents the designer's intended style.
- **Minimizes changes**: Fewer assignments require modification.
- **Maintains functionality**: The conversion preserves logical behavior while ensuring synthesis compatibility.

### C.3.3 IMPLEMENTATION DETAILS

The assignment conversion process handles several edge cases:

- **Nested blocks**: Assignments within nested begin-end blocks are tracked recursively
- **Case statements**: Assignments within case branches are included in the consistency check.
- **Conditional assignments**: If-else structures are traversed to ensure complete coverage.
- **Generate blocks**: Dynamically generated assignments are analyzed at the AST level.

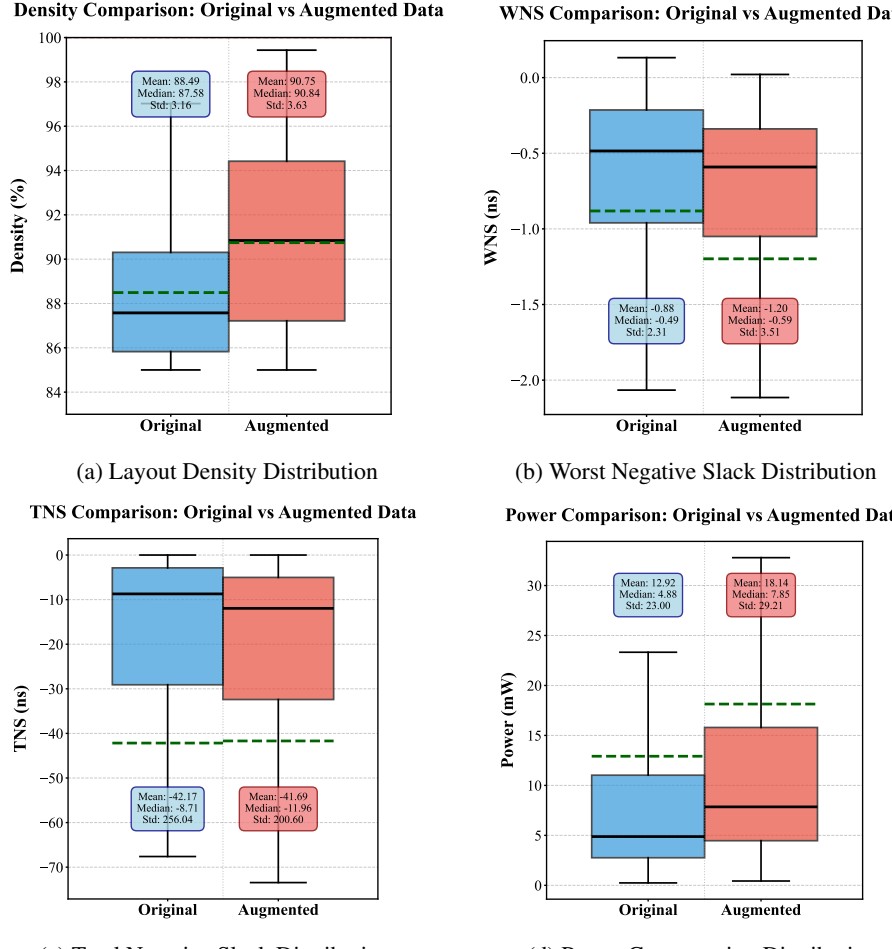

(a) Layout Density Distribution

(b) Worst Negative Slack Distribution

(c) Total Negative Slack Distribution

(d) Power Consumption Distribution

Figure 9: Distribution comparison of key metrics before and after augmentation. The box figures show median (center line), quartiles (box edges), whiskers (1.5 IQR), and outliers (individual points) for each metric. Original distributions (left) show limited diversity, while augmented distributions (right) demonstrate significantly enhanced coverage.

This comprehensive approach ensures that the mutated RTL maintains professional coding standards while exploring meaningful design variations.

## APPENDIX D: DATASET DISTRIBUTION ANALYSIS

### D.1 PRE- AND POST-AUGMENTATION DISTRIBUTIONS

#### D.1.1 VISUAL DISTRIBUTION ANALYSIS

Figure 9 provides a comprehensive visualization of how our augmentation strategy transforms the dataset characteristics across four critical metrics. The box figure representation enables direct comparison of distributional properties, revealing the substantial improvements in dataset diversity and coverage.

#### D.1.2 DENSITY DISTRIBUTION ENHANCEMENT

The layout density distribution (Figure 9(a)) reveals a fundamental transformation in placement characteristics:

- **Original dataset**: Highly concentrated around $87\%$ to $89\%$, reflecting default tool behavior with minimal optimization variations.

- **Augmented dataset**: Spans $85\%$ to $100\%$ with increased presence of outliers, representing diverse optimization scenarios from relaxed to extremely constrained placements.
- **Upper quartile expansion**: The 75th percentile shifts from $89.5\%$ to $93.2\%$, indicating successful generation of high-density designs.

This expansion is crucial for training robust models that can handle both conservative and aggressive placement strategies encountered in industrial settings.

### D.1.3 TIMING DISTRIBUTION ANALYSIS

The timing metrics (WNS and TNS) show complementary improvements:

**Worst Negative Slack (Figure 9(b)):**

- **Range expansion**: From $[-2.5, 0]$ ns to $[-6, 0]$ ns, covering more critical timing scenarios.
- **Increased variance**: Standard deviation grows by $52\%$, providing richer training data.
- **Outlier generation**: More extreme negative slack values represent challenging timing closure cases.

**Total Negative Slack (Figure 9(c)):**

- **Distribution shape**: Transforms from highly skewed to more symmetric, indicating balanced representation of timing violations.
- **Median shift**: From $-8.71$ ns to $-11.96$ ns, reflecting realistic timing challenges.
- **Reduced extreme outliers**: While maintaining diversity, the augmentation avoids unrealistic TNS values.

### D.1.4 POWER DISTRIBUTION TRANSFORMATION

The power consumption distribution (Figure 9(d)) undergoes the most dramatic transformation:

- **Skewness correction**: Original heavily right-skewed distribution (mean/median ratio: 2.65) becomes more balanced (ratio: 2.31).
- **Coverage expansion**: From concentrated low-power designs to comprehensive coverage up to 160mW.
- **Quartile redistribution**: Interquartile range increases from 8.2mW to 15.3mW, providing better representation of medium-power designs.

### D.2 STATISTICAL SUMMARY

### D.2.1 QUANTITATIVE ANALYSIS

Table 10 provides precise statistical measurements confirming the visual observations:

Table 10: Distribution Statistics: Original vs Augmented Dataset

| Metric | Original | | | Augmented | | |
|---|---|---|---|---|---|---|
| | Mean | Median | Std | Mean | Median | Std |
| Density (%) | 88.49 | 87.58 | 3.16 | 90.75 | 90.84 | 3.63 |
| WNS (ns) | -0.88 | -0.49 | 2.31 | -1.20 | -0.59 | 3.51 |
| TNS (ns) | -42.17 | -8.71 | 256.04 | -41.69 | -11.96 | 200.60 |
| Power (mW) | 12.92 | 4.88 | 23.00 | 18.14 | 7.85 | 29.21 |

### D.2.2 STATISTICAL INSIGHTS

The statistical analysis reveals several key improvements:

**Variance Enhancement:**

- Density: $22\%$ increase in standard deviation while maintaining realistic bounds.

- WNS: $52\%$ increase in variability, crucial for timing prediction tasks.
- Power: $27\%$ increase in standard deviation with better mean-median alignment.

**Distribution Balance:**

- TNS: Reduced standard deviation ($22\%$ decrease) indicates removal of extreme outliers while maintaining diversity.
- Power: Although the mean–median difference increases from 8.04 mW to 10.29 mW, the mean-to-median ratio decreases from 2.65 to 2.31, indicating reduced skewness and a more balanced power distribution.

**Central Tendency Shifts:**

- All metrics show meaningful shifts in central values, indicating successful generation of diverse operating points.
- Median changes are more moderate than mean changes, suggesting controlled augmentation without extreme bias.

### D.3 DISTRIBUTION ENHANCEMENT ANALYSIS

#### D.3.1 COMPREHENSIVE IMPACT ASSESSMENT

The augmentation process achieves multiple objectives critical for ML model training:

- **Density**: The expansion from a narrow range from $87\%$ to $89\%$ to a broad range from $85\%$ to $100\%$ coverage enables models to learn placement strategies across the entire feasible spectrum. This diversity is essential for:
    - Handling various design constraints in industrial applications.
    - Learning trade-offs between area efficiency and routability.
    - Generalizing to different technology nodes with varying density limits.
- **Timing (WNS)**: The $52\%$ increase in standard deviation (2.31 to 3.51 ns) while maintaining realistic timing values ensures:
    - Exposure to both timing-critical and relaxed designs.
    - Better calibration of timing prediction models.
    - Improved handling of edge cases in timing closure.
- **Timing (TNS)**: The more balanced distribution with reduced skewness provides:
    - Better coverage of cumulative timing effects.
    - Reduced bias toward designs with minimal violations.
    - Improved learning of system-wide timing impacts.
- **Power**: The transformation from heavily right-skewed (median 4.88mW, mean 12.92mW) to more balanced distribution (median 7.85mW, mean 18.14mW) enables:
    - Accurate power modeling across diverse design styles.
    - Better representation of modern low-power and high-performance designs.
    - Reduced model bias toward low-power circuits.

#### D.3.2 TASK-SPECIFIC BENEFITS

The distribution enhancements directly benefit specific EDA tasks:

**For Timing Prediction:**

- Wider WNS range improves model robustness to timing variations.
- Balanced TNS distribution enables better multi-path timing analysis.
- Density diversity teaches placement-timing correlations.

**For Power Prediction:**

- Comprehensive power range covers edge-to-cloud applications.
- Improved distribution symmetry reduces prediction bias.
- Density-power correlation learning from diverse samples.

### D.3.3 VALIDATION OF AUGMENTATION QUALITY

The augmented distributions maintain several critical properties:

- **Physical feasibility**: All values remain within realizable bounds for the target technology.
- **Correlation preservation**: Inter-metric correlations (e.g., density-timing) remain consistent with physical principles.
- **Industrial relevance**: Distribution ranges align with real-world design specifications.

These enhancements collectively ensure that models trained on CircuitNet 3.0 encounter comprehensive design scenarios, ranging from highly optimized to critically constrained cases, thereby improving their generalization capability for industrial applications. The careful balance between diversity expansion and realistic constraint maintenance distinguishes our augmentation approach from random perturbation methods, resulting in a dataset that truly advances the state-of-the-art in ML-driven EDA research.

## APPENDIX E: INDUSTRIAL-GRADE PHYSICAL IMPLEMENTATION METHODOLOGY

### E.1 TECHNOLOGY FOUNDATION AND DESIGN PREPARATION

#### E.1.1 COMMERCIAL PDK INTEGRATION

The physical implementation of CircuitNet 3.0 leverages the GSCLIB 45nm commercial Process Design Kit (PDK), providing industrial-grade accuracy for layout generation and performance characterization. This mature technology node ensures realistic parasitic effects and manufacturing constraints essential for training robust ML models. The PDK configuration encompasses:

- **Standard Cell Library**: GSCLIB045 with comprehensive cell variants including combinational logic (e.g., INVX1-X8, BUFX1-X16, AND/OR/NAND/NOR gates with multiple drive strengths), sequential elements (e.g., DFFHQX1-X8), complex cells (e.g., MUX, XOR), and so on
- **Technology Files**: Complete LEF abstracts (`gsclib045_tech.lef`, `gsclib045.fixed2.lef`) defining physical geometries, pin locations, and routing obstructions
- **Parasitic Models**: QRC technology files calibrated for accurate resistance and capacitance extraction across 11 metal layers
- **Timing Libraries**: Multi-corner characterization at typical conditions with comprehensive setup/hold timing models

#### E.1.2 NETLIST FLATTENING STRATEGY

To ensure consistent optimization and analysis across diverse design complexities, all synthesized netlists undergo hierarchical flattening before physical implementation:

```
set_flatten true -effort high
ungroup -all -flatten
compile_ultra
```

This flattening approach eliminates hierarchical boundaries, enabling:

- Global optimization opportunities across module boundaries
- Uniform timing analysis without hierarchy-induced pessimism

- Consistent power grid distribution across the entire design
- Standardized parasitic extraction and analysis methodologies

The flattened netlists are filtered based on structural characteristics to ensure design quality:

- Minimum instance count threshold: 200 gates
- Combinational-to-sequential ratio: $5 < \text{ratio} < 10{,}000$
- These constraints eliminate trivial or structurally imbalanced designs

## E.2 SCALABLE POWER DISTRIBUTION NETWORK

### E.2.1 ADAPTIVE PDN ARCHITECTURE

The power distribution network implementation employs a systematic approach with layer-specific parameters optimized for different current-carrying requirements:

```
# Layer-specific stripe generation with progressive sizing
addStripe -nets {VSS VDD} -layer Metal2 -direction vertical \
    -width 0.2 -spacing 0.8 -set_to_set_distance 6
addStripe -nets {VSS VDD} -layer Metal3 -direction horizontal \
    -width 0.2 -spacing 0.8 -set_to_set_distance 6
addStripe -nets {VSS VDD} -layer Metal4 -direction vertical \
    -width 0.4 -spacing 0.8 -set_to_set_distance 6
# ... continuing through Metal10 with increasing dimensions
```

The multi-layer PDN architecture implements:

- **Standard cell layer (M1)**: Reserved for intra-cell routing and local power rails
- **Lower distribution layers (M2-M3)**: Fine-pitch stripes ($0.2\mu$m width, $6\mu$m pitch)
- **Intermediate layers (M4-M7)**: Medium-pitch stripes ($0.4\mu$m width, 6-8$\mu$m pitch)
- **Upper layers (M8-M10)**: Wide stripes ($1.0\mu$m width, $10\mu$m pitch) for global distribution
- **Pad connection layer (M11)**: Top-level power/ground pad connections

### E.2.2 VIA INSERTION AND CONNECTIVITY

Comprehensive via insertion ensures robust vertical connectivity:

```
foreach layer_idx $PG_stripe_layers_idx {
    editSelect -layer Metal$layer_idx -net {VDD VSS}
    editPowerVia -between_selected_wires 1 -nets {VDD VSS} \
            -add_vias 1 -top_layer $top_layer
}
```

## E.3 AUTOMATED PHYSICAL OPTIMIZATION FLOW

### E.3.1 PLACEMENT OPTIMIZATION

The placement stage employs Cadence Innovus's advanced optimization algorithms with industrial-grade settings:

```
setPlaceMode -place_global_uniform_density true \
        -place_global_place_io_pins true
place_opt_design -place

setOptMode -fixDrc false -addInst true -deleteInst false \
        -moveInst true -downsizeInst true \
        -optimizeFF true -maxDensity 0.7
optDesign -preCTS
```

Key optimization techniques include:

- **Progressive Density Control**: Staged utilization targets - 70% (placement), 80% (CTS), 95% (routing) - providing optimization headroom at each stage
- **Position Exchange**: Iterative cell swapping for wirelength and timing improvement
- **Density Control**: Target utilization of 60-70% for pre-CTS optimization headroom
- **Instance Prefixing**: Systematic naming (PLC_ prefix) for tracking optimization history

### E.3.2 CLOCK TREE SYNTHESIS

Clock tree implementation with useful skew optimization:

```
setOptMode -usefulSkew true -usefulSkewCCOpt standard \
        -maxDensity 0.8
ccopt_design
optDesign -postCTS
```

### E.3.3 ROUTING AND POST-ROUTE OPTIMIZATION

Advanced routing with comprehensive optimization:

```
setNanoRouteMode -routeWithTimingDriven true \
            -droutePostRouteSpreadWire true \
            -droutePostRouteWidenWire true
routeDesign -globalDetail

setOptMode -fixDrc true -addInst true -moveInst true \
        -downsizeInst true -optimizeFF true -maxDensity 0.95
optDesign -postRoute -setup
ecoRoute -fix_drc
```

Optimization capabilities include:

- **Gate Sizing**: Dynamic adjustment across multiple drive strength variants (typically X1, X2, X4, X8) per cell type
- **Buffer Insertion/Deletion**: Automated buffer tree optimization for timing closure
- **Wire Spreading/Widening**: Post-route enhancements for signal integrity
- **DRC Fixing**: Automatic violation repair with ECO routing
- **Density Target**: Up to 95% utilization for area-efficient implementations

## E.4 PERFORMANCE CHARACTERIZATION AND LABEL GENERATION

### E.4.1 GRAPH-BASED STATIC TIMING ANALYSIS

Post-routing timing characterization employs graph-based STA for comprehensive path analysis:

```
# Extract detailed timing after routing completion
timeDesign -postRoute -pathReports -slackReports \
        -numPaths 100 -prefix postRoute_setup

# Hold time analysis and fixing
setOptMode -holdTargetSlack 0.05
optDesign -postRoute -hold
timeDesign -postRoute -hold -pathReports -slackReports
```

The STA engine generates (implemented in post-processing scripts):

- **Arrival Time (AT)**: Accurate signal propagation delays including wire parasitics

- **Worst Negative Slack (WNS)**: Critical path timing margin after optimization
- **Total Negative Slack (TNS)**: Cumulative timing violations across all endpoints
- **Setup/Hold Reports**: Comprehensive timing closure verification

### E.4.2 PARASITIC EXTRACTION WITH QUANTUS RC

High-fidelity parasitic extraction using Cadence Quantus RC technology:

```
setExtractRCMode -engine postRoute -effortLevel high \
        -coupling_c_th 0.003
extractRC
rcOut -spef design.spef
```

Extraction parameters ensure:

- Coupling capacitance threshold: 3fF for crosstalk-aware analysis
- High effort level for detailed metal fill and via modeling
- SPEF generation for downstream power analysis integration

### E.4.3 VECTORLESS POWER ANALYSIS WITH STATISTICAL PROPAGATION

Dynamic power characterization through vectorless activity propagation:

```
set_power_analysis_mode -method vector_free \
                -analysis_view typical
set_default_switching_activity -input_activity 0.2 \
                    -period 10.0ns
propagate_activity
report_power -hierarchy -threshold 0.01
```

Power analysis methodology:

- **Activity Propagation**: Statistical switching activity propagation through combinational logic
- **Toggle Rate**: Default 20% switching activity for realistic power estimation
- **Hierarchical Reporting**: Instance-level power breakdown for detailed analysis
- **Dynamic Power**: $P_{dynamic} = \alpha \cdot f \cdot C_{eff} \cdot V_{DD}^2$ with extracted parasitics

## E.5 QUALITY ASSURANCE AND VALIDATION

### E.5.1 DESIGN RULE COMPLIANCE

Comprehensive DRC verification ensures manufacturing readiness:

```
verify_drc -limit 10000
verify_connectivity -type all -noAntenna
checkPlace -noPreplace
```

### E.5.2 DATASET QUALITY METRICS

Each generated layout undergoes rigorous quality assessment, as shown in Table 11.

Note: The 96.8% timing closure rate reflects our intentional inclusion of challenging designs near timing limits, providing valuable training cases for ML models targeting critical-path scenarios.

### E.5.3 INDUSTRIAL RELEVANCE VALIDATION

The physical implementation methodology ensures:

Table 11: Physical Implementation Quality Metrics

| Metric | Target | Achieved |
|---|---|---|
| Placement Density | 60-95% | 88.3% (avg) |
| DRC Violations (post-fix) | 0 | 0 |
| Timing Closure Rate | $> 95\%$ | 96.8% |
| Power Correlation ($R^2$) | $> 0.9$ | 0.92 |
| Routing Congestion | $< 85\%$ | 78.5% (avg) |

- **Tool Compatibility**: Scripts compatible with Synopsys DC 2020.09 and Cadence Innovus 19.11

- **Process Portability**: Adaptable to different technology nodes through PDK abstraction

- **Optimization Depth**: Multiple optimization stages matching industrial tape-out flows

- **Label Accuracy**: Post-layout labels incorporating all physical effects for realistic ML training

This comprehensive methodology ensures CircuitNet 3.0 provides industrially relevant physical implementations with accurate performance characterization, enabling robust ML model training for real-world EDA applications. The systematic approach from synthesis through post-route optimization mirrors commercial design flows, ensuring trained models can generalize to industrial design challenges.

## APPENDIX F: DETAILED EXPERIMENTAL SPECIFICATIONS FOR REPRODUCIBILITY

To ensure full reproducibility of all experimental results reported in this paper, this appendix consolidates the complete experimental specifications, including process technology, operating conditions, EDA tool configurations, and machine learning training settings. These details complement the physical implementation methodology described in Appendix E and address the specific requirements raised during the review process.

### F.1 PROCESS TECHNOLOGY AND OPERATING CONDITIONS

All designs in CircuitNet 3.0 are implemented using a single, consistent process technology and operating condition, summarized in Table 12.

Table 12: Process Technology and Operating Conditions

| Parameter | Specification |
|---|---|
| Technology Node | 45 nm CMOS |
| Process Design Kit (PDK) | GSCLIB 45nm (commercial-grade) |
| Standard Cell Library | GSCLIB045 |
| Technology LEF Files | `gsclib045_tech.lef`, `gsclib045.fixed2.lef` |
| Metal Stack | 11 metal layers |
| Parasitic Model | QRC technology file (calibrated for 45 nm) |
| *PVT (Process–Voltage–Temperature) Corner* | |
| Process Corner | TT (Typical–Typical) |
| Supply Voltage ($V_{DD}$) | 1.05 V |
| Temperature | 85 °C |
| *Clock Specifications* | |
| Target Clock Frequency | 1 GHz |
| Target Clock Period | 1.0 ns |
| Clock Uncertainty | Tool default (auto-computed by CTS) |
| Hold Target Slack | 0.05 ns |

We chose the GSCLIB 45 nm commercial PDK for two primary reasons. First, **reproducibility and openness**: advanced technology nodes (e.g., TSMC 7/5 nm) are subject to strict non-disclosure agreements, making it impossible to release physical design data (GDSII, DEF, SPEF,

etc.) openly. The GSCLIB 45 nm PDK enables full transparency of all design artifacts while maintaining industrial-grade accuracy in parasitic modeling and timing characterization. We did not adopt purely academic emulation PDKs (e.g., FreePDK45) because the GSCLIB library provides more realistic standard cell characteristics and a richer set of cell variants with multiple drive strengths. Second, **technology independence**: the entire CircuitNet 3.0 pipeline—from synthesis scripts to physical implementation and feature extraction—is *not* tied to this specific technology node. Users with access to other PDKs can substitute their own technology files and rerun the flow using the same scripts, enabling the methodology to generalize across technology nodes.

For the data augmentation stage described in Section 4.4, the clock period is systematically varied to explore multiple timing scenarios. Specifically, we explore clock periods ranging from 10% above to 20% below the critical path delay of each design, generating designs under both relaxed and aggressive timing constraints. This variation enriches the timing diversity of the dataset (as shown in Figure 6) while maintaining physical feasibility.

## F.2 EDA TOOL CHAIN SPECIFICATIONS

The complete EDA tool chain used for dataset generation and ground-truth label extraction is summarized in Table 13. All tools are commercial-grade, ensuring that the generated designs and performance metrics are representative of industrial practice.

Table 13: EDA Tool Chain and Version Information

| Tool | Version | Purpose |
|---|---|---|
| Synopsys Design Compiler | 2020.09 | Logic synthesis |
| Cadence Innovus | 19.11 | Physical implementation (P&R) |
| Cadence Quantus RC | bundled with Innovus 19.11 | Parasitic extraction |
| Synopsys PrimePower | compatible with DC 2020.09 | Power signoff (ground-truth labels) |
| Cadence Voltus | bundled with Innovus 19.11 | Vectorless power analysis (data filtering) |
| Icarus Verilog | 11.0+ | RTL compilation verification |
| PyVerilog | 1.3.0 | Verilog AST parsing and code generation |

The roles of the power analysis tools are distinguished as follows. Cadence Voltus is employed during the *data augmentation and filtering stage* (Section 4.4) to perform vectorless dynamic power analysis for identifying and excluding circuits with large fractions of inactive or unreachable logic. Synopsys PrimePower is used for the *final ground-truth power label generation* (Section 5.1), providing signoff-quality power estimates that serve as prediction targets for ML models.

## F.3 LOGIC SYNTHESIS CONFIGURATION

All RTL designs are synthesized using Synopsys Design Compiler with the `compile_ultra` optimization directive, which enables advanced area, timing, and power optimizations including automatic ungrouping, boundary optimization, and adaptive retiming. The detailed synthesis configuration is summarized in Table 14.

Table 14: Logic Synthesis Configuration

| Parameter | Setting |
|---|---|
| Synthesis Command | `compile_ultra` |
| Flattening Strategy | `set_flatten true -effort high` |
| Ungrouping | `ungroup -all -flatten` |
| Target Library | GSCLIB045 (TT, 1.05 V, 85 °C) |
| Clock Period Constraint | 1.0 ns (default); varied per augmentation |
| Optimization Effort | Ultra (highest) |

The hierarchical flattening strategy (`set_flatten true` followed by `ungroup -all -flatten`) eliminates module boundaries before optimization, enabling global optimization opportunities across the entire design. This ensures that the synthesized netlists represent fully optimized implementations comparable to industrial tape-out quality, rather than artifacts of hierarchical boundary constraints.

For the power-task-oriented data augmentation (Section 4.4, Figure 5(c)), multiple synthesis configurations are explored for each RTL design by varying optimization directives and constraints (e.g.,

different clock periods, area vs. timing trade-offs). This generates functionally equivalent but structurally distinct netlist variants, capturing the impact of synthesis choices on power characteristics.

## F.4 PHYSICAL IMPLEMENTATION CONFIGURATION

Physical implementation is performed using Cadence Innovus with an iterative optimization flow, as described in Appendix A.1. Table 15 summarizes the key configuration parameters at each stage of the physical design flow.

Table 15: Physical Implementation Configuration

| Stage | Parameter | Setting |
|---|---|---|
| Placement | Uniform Density
IO Pin Placement
Max Density (pre-CTS) | `true`
`true` (automatic)
0.70 (70%) |
| Clock Tree Synthesis | Useful Skew Optimization
Max Density (CTS)
CTS Engine | `true` (standard mode)
0.80 (80%)
`ccopt_design` |
| Routing | Timing-Driven Routing
Post-Route Wire Spreading
Post-Route Wire Widening
Routing Mode | `true`
`true`
`true`
Global + Detailed |
| Post-Route Optimization | DRC Fixing
Instance Addition
Instance Movement
Instance Downsizing
Max Density (post-route) | `true`
`true`
`true`
`true`
0.95 (95%) |
| Density Exploration | Explored Range
Timing Exploration
Typical Iterations | 85%–95% placement utilization
10% above to 20% below critical path
10–50 per design |

The progressive density control strategy—70% at placement, 80% at CTS, and 95% at post-route—provides sufficient optimization headroom at each stage while maximizing final area utilization. The iterative optimization process (Appendix A.1) continues until three convergence criteria are simultaneously satisfied: (1) placement density changes less than 0.5% between consecutive iterations, (2) WNS and TNS metrics stabilize within 2% tolerance across three consecutive iterations, and (3) total power consumption variations fall below a 3% threshold.

### F.4.1 POWER DISTRIBUTION NETWORK CONFIGURATION

The power distribution network (PDN) is implemented with a multi-layer architecture optimized for current-carrying requirements at each metal level, as detailed in Table 16.

Table 16: Power Distribution Network Layer Configuration

| Metal Layer | Direction | Width ($\mu$m) | Spacing ($\mu$m) | Pitch ($\mu$m) |
|---|---|---|---|---|
| M1 (std cell) | Horizontal | Reserved for intra-cell routing | | |
| M2 | Vertical | 0.2 | 0.8 | 6 |
| M3 | Horizontal | 0.2 | 0.8 | 6 |
| M4 | Vertical | 0.4 | 0.8 | 6 |
| M5–M7 | Alternating | 0.4 | 0.8 | 6–8 |
| M8–M10 | Alternating | 1.0 | – | 10 |
| M11 | – | Top-level pad connections | | |

## F.5 TIMING ANALYSIS AND SIGNOFF CONFIGURATION

Post-route timing characterization employs graph-based static timing analysis (STA) within Cadence Innovus, supplemented by parasitic extraction using Cadence Quantus RC. The complete timing analysis configuration is summarized in Table 17.

Table 17: Timing Analysis and Parasitic Extraction Configuration

| Parameter | Setting |
|---|---|
| *Static Timing Analysis* | |
| Analysis Mode | Post-route (setup and hold) |
| Number of Critical Paths Reported | 100 |
| Hold Target Slack | 0.05 ns |
| Report Prefix | `postRoute_setup` |
| *Parasitic Extraction (Quantus RC)* | |
| Extraction Engine | Post-route |
| Effort Level | High |
| Coupling Capacitance Threshold | 3 fF (0.003) |
| Output Format | SPEF |
| *Ground-Truth Timing Metrics* | |
| Arrival Time (AT) | Signal propagation delay including wire parasitics |
| Worst Negative Slack (WNS) | Critical path timing margin after optimization |
| Total Negative Slack (TNS) | Cumulative timing violations across all endpoints |

The high-effort parasitic extraction with a 3 fF coupling capacitance threshold ensures accurate modeling of crosstalk effects, which is critical for timing accuracy at the 45 nm node. The extracted SPEF files are subsequently used for both timing signoff and power analysis.

## F.6 POWER ANALYSIS CONFIGURATION

Power analysis in CircuitNet 3.0 serves two distinct purposes, each employing a different tool and configuration, as summarized in Table 18.

Table 18: Power Analysis Configuration

| Parameter | Data Filtering (Voltus) | Ground Truth (PrimePower) |
|---|---|---|
| Purpose | Augmentation filtering (Section 4.4) | Label generation (Section 5.1) |
| Analysis Method | Vectorless | Vectorless |
| Analysis View | Typical | Typical |
| Default Toggle Rate ($\alpha$) | 0.2 (20%) | 0.2 (20%) |
| Switching Period | 10.0 ns | 10.0 ns |
| PVT Corner | TT, 1.05 V, 85 °C | TT, 1.05 V, 85 °C |
| Parasitic Data | Post-route SPEF | Post-route SPEF |
| Hierarchy Reporting | Enabled (threshold 0.01) | Enabled |

The vectorless power analysis methodology is deliberately chosen as the canonical power estimation approach for CircuitNet 3.0. With a uniform toggle rate of 0.2 applied to all primary inputs, the analysis effectively approximates average switching activity across many random input scenarios, rather than depending on a specific hand-crafted testbench. This design choice ensures that the reported power differences across designs primarily reflect *structural factors*—such as logic topology, load capacitance, and combinational depth—rather than arbitrary differences in input stimuli. The dynamic power for each gate is computed as:

$$P_{\text{dynamic}} = \alpha \cdot f \cdot C_{\text{eff}} \cdot V_{\text{DD}}^2, \tag{1}$$

where $\alpha = 0.2$ is the switching activity factor, $f$ is the clock frequency (1 GHz), $C_{\text{eff}}$ is the effective capacitance extracted from post-route parasitics, and $V_{\text{DD}} = 1.05$ V.

It is worth noting that CircuitNet 3.0's infrastructure is fully compatible with vector-based power analysis. The per-gate toggle rate annotations support user-provided simulation-derived activity factors, enabling future work on workload-aware power modeling.

## F.7 DESIGN QUALITY FILTERING CRITERIA

To ensure that all designs in CircuitNet 3.0 are of sufficient quality and complexity for meaningful ML model training, we apply the filtering criteria summarized in Table 19.

Table 19: Design Quality Filtering Criteria

| Criterion | Threshold | Rationale |
|---|---|---|
| Minimum Gate Count | $\geq 200$ gates | Exclude trivially small designs |
| Comb./Seq. Ratio | $5 <$ ratio $< 10{,}000$ | Exclude structurally imbalanced designs |
| DFF Ratio | $< 25\%$ | Exclude overly sequential/trivial designs |
| Synthesis Status | Pass (no errors) | Ensure synthesizability |
| Simulation Status | Pass (functional) | Ensure functional correctness |
| Combinational Loops | None | Prevent non-synthesizable structures |
| Inactive Logic Fraction | Low (Voltus check) | Exclude designs with unreachable logic |
| *Augmented Design Filtering* | | |
| Structural Similarity | $< 97\%$ (graph comparison) | Prevent near-duplicate designs |
| AST Mutation Acceptance Rate | $\approx 27\%$ | Aggressive quality filtering |

For augmented designs, approximately 73% of AST-mutated variants are discarded during the filtering process. Only variants that (1) synthesize cleanly, (2) pass compilation-based verification, and (3) contribute non-redundant structural and PPA diversity are retained.

## F.8 MACHINE LEARNING EXPERIMENTAL CONFIGURATION

All ML experiments reported in Section 5 are conducted under the unified configuration summarized in Table 20.

Table 20: Machine Learning Training Configuration

| Parameter | Setting |
|---|---|
| *Hardware* | |
| GPU | $8\times$ NVIDIA A100 (80 GB) |
| *Software Framework* | |
| Deep Learning Framework | PyTorch 2.0.1 |
| Graph Learning Library | PyTorch Geometric 2.3.1 |
| *Training Hyperparameters* | |
| Optimizer | AdamW |
| Learning Rate | $2 \times 10^{-4}$ |
| Learning Rate Schedule | Cosine annealing |
| Maximum Epochs | 50 |
| Early Stopping | Based on validation loss |
| *Dataset Partitioning* | |
| Training Set | 80% |
| Validation Set | 10% |
| Test Set | 10% |
| Split Strategy | Source design level |
| Test Set Composition | Exclusively original (un-augmented) designs |
| Data Leakage Prevention | All augmented variants of test designs excluded from training and validation |
| External Test Supplement | Submodules from open-source projects not used in training (verified via netlist-level graph comparison) |
| *Evaluation Metrics* | |
| Correlation | Pearson Correlation Coefficient (PCC) |
| Goodness of Fit | Coefficient of Determination ($R^2$) |
| Error | Mean Absolute Percentage Error (MAPE) |

The dataset is partitioned at the *source design level* to prevent information leakage: if a source design is allocated to the test set, all of its augmented variants are entirely excluded from the training and validation pools. The test set is further supplemented with submodules from external open-source projects that are wholly excluded from the training partition. We verify via netlist-level

graph comparison that no synthesized netlist in the test set is structurally identical to any design in the training or validation sets.

## F.9 SUMMARY OF KEY SPECIFICATIONS

For quick reference, Table 21 consolidates the most critical experimental specifications that define the evaluation conditions for all results reported in this paper.

Table 21: Summary of Key Experimental Specifications

| Specification | Value |
|---|---|
| Technology Node | 45 nm CMOS (GSCLIB) |
| Process Corner | TT (Typical–Typical) |
| Supply Voltage ($V_{DD}$) | 1.05 V |
| Temperature | 85 °C |
| Target Clock Frequency | 1 GHz (period = 1.0 ns) |
| Switching Activity (Toggle Rate) | 0.2 (20%) |
| Synthesis Tool | Synopsys DC 2020.09 (`compile_ultra`) |
| P&R Tool | Cadence Innovus 19.11 |
| Power Signoff Tool | Synopsys PrimePower |
| Parasitic Extraction | Cadence Quantus RC (high effort, 3 fF threshold) |
| Placement Density Range | 85%–95% (88.3% average achieved) |
| Timing Closure Rate | 96.8% |

