# OpenReview forum: "CircuitNet 3.0: A Multi-Modal Dataset with Task-Oriented Augmentation for AI-Driven Circuit Design"
_ICLR.cc/2026/Conference — ICLR 2026 Poster_

### Official Review · Reviewer_myKG · 2025-10-27

**Soundness:** 3
**Presentation:** 3
**Contribution:** 3
**Rating:** 6
**Confidence:** 3

**Summary:**

The paper presents CircuitNet 3.0, a benchmark dataset designed to support AI-driven EDA research. It introduces a multi-modal framework that integrates RTL, gate-level netlist, and layout representations, enabling cross-stage learning for circuit-level prediction tasks such as timing and power estimation.

**Strengths:**

The paper presents a well-structured and comprehensive effort to advance dataset resources for AI-driven EDA research. Its originality lies in integrating multiple design representations (RTL, gate-level netlist, and layout) into a unified multi-modal dataset, and provides a new level of data granularity for model training. The proposed task-oriented augmentation framework demonstrates an application of AST-based code mutation in the EDA context, contributing to the diversity and robustness of learning samples.

**Weaknesses:**

1. All reported improvements are based on models trained with augmented datasets that are larger than the original data or baseline benchmark (Section 4.4 and Table 5–6). The paper lacks ablation or control experiment to verify that the observed gains stem from the augmentation methodology rather than simply from increased volume of the training set.
2. For the timing-prediction task (Table 4a), the authors attribute performance gains of RTLDistil to multi-modal learning, yet the compared non-multi-modal baselines (MasterRTL, RTL-Timer) may differ substantially in architecture, capacity, and parameter count. There lacks controlled "with/without multi-modal input" experiment for timing.
Similarly, for MOSS in the power-prediction task (Table 4b), the attribution of performance gain solely to "multi-modal representation" is not fully justified without evaluating RTL-only, netlist-only, and combined-input variants under identical configurations.
3. CircuitNet 3.0 is positioned as a task-oriented benchmark, yet the tasks are limited to timing and power prediction. Other critical circuit-design objectives—e.g., routability and IR-drop mentioned in CircuitNet 2.0—are absent. By focusing narrowly on two metrics, the dataset may bias models toward timing/power trade-offs at the expense of broader PPA generalization.

**Questions:**

1. The claimed 3.95% error reduction in power prediction (e.g., line 92, 467 and 485) lacks exact calculation and appears inconsistent with MAPE reductions between 11–23% shown in Table 6.

2. The paper highlights "a novel data augmentation framework based on Verilog abstract syntax trees (ASTs)" (Section 4.3) as a core contribution. However, AST-guided code mutation and transformation have been widely explored in software-engineering contexts [1]. So it would be better if the authors could elaborate on how the proposed AST-based augmentation differs technically from prior AST mutation and program-transformation works.

3. The paper states (line 357) that the test set was 'supplemented with submodules from external open-source projects,' but does not justify the need for this supplementation or provide details on the data's origin. This omission raises concerns about potential overlap between the training and test data, especially since section A.2 lists GitHub, Hugging Face, RISC-V, and OpenCores as sources for both training and validation data

4. In Figure 6(b) and 6(d), both the post-augmentation WNS and total-power distributions exhibit an unusual concentration of samples around layout-density ≈ 94.5%, which is not present in the pre-augmentation distribution shown in Figure 6(a) and 6(c). Could the authors clarify:
(1) what specific synthesis or placement-density configurations led to this clustering?
   (2) why this effect appears near the 94.5% density rather than being smoothly distributed?
   (3) whether such clustering introduces dataset bias that may influence model evaluation or overfit models toward a narrow design-density region?

5. It would be beneficial for the authors to discuss how this work relates to recent benchmarks such as ChiPBench [2], which also targets AI-based EDA evaluation with a complete open-source flow.

[1] Petrović, Goran, et al. "Practical mutation testing at scale: A view from google." IEEE Transactions on Software Engineering 48.10 (2021): 3900-3912.

[2] Wang, Zhihai, et al. Benchmarking end-to-end performance of ai-based chip placement algorithms. The Thirty-ninth Annual Conference on Neural Information Processing Systems Datasets and Benchmarks Track.

---

> ### Author Response · Authors · 2025-11-26
> **Response to Reviewer myKG's weakness 1**
>
> ### Response to Reviewer myKG
>
> We thank Reviewer **myKG** for the careful review and constructive suggestions. Below we address each weakness and question in turn. We hope to receive your approval and look forward to further communication with you.
>
> ---
>
> #### Weakness 1 – Are gains due to augmentation *content* or just more data?
>
> **Comment.**
> All reported improvements use the augmented training set, which is larger than the original or baseline datasets (Sec. 4.4, Tables 5–6). There are no ablations to show whether the gains come from the augmentation method itself or simply from having more training examples.
>
> **Answer.**
> This is an important point. We conducted additional experiments during the rebuttal to disentangle “more data” from “better (task-oriented) data.”
>
> First, note that Fig. 6 shows a **substantial widening of the WNS distribution** after AST-based augmentation: in the original dataset, most designs have WNS close to 0 ns (timing met), so the model sees very few “violating” hard cases. After augmentation, WNS expands roughly to [-6 ns, 0 ns], introducing many samples with significant timing violations. Similarly, power coverage extends to higher ranges. This indicates that our augmentation is injecting **new hard examples**, not just copies.
>
> To explicitly control for dataset size, we performed a “data doubling” experiment:
>
> * **Original**: original training set.
> * **Original (data×2)**: the original training set duplicated (randomly shuffled) so that the number of training samples is *slightly larger* than in the augmented dataset, but **without any augmentation**.
> * **Augmented (ours)**: our full task-oriented augmented dataset.
>
> All models and training settings are kept identical across these three variants.
>
> The results for **RTLDistil on timing prediction** are:
>
>
> Table A: RTLDistil on original vs. doubled vs. augmented data
>
> | Dataset              | AT PCC | AT R² | AT MAPE↓ | WNS PCC | WNS R² | WNS MAPE↓ | TNS PCC | TNS R² | TNS MAPE↓ |
> |----------------------|:------:|:-----:|:--------:|:-------:|:------:|:---------:|:-------:|:------:|:---------:|
> | Original             | 0.887  | 0.785 | 19.72%   | 0.871   | 0.756  | 35.28%    | 0.918   | 0.841 | 40.15%    |
> | Original (data×2)    | 0.879  | 0.775 | 19.11%   | 0.882   | 0.747  | 33.79%    | 0.909   | 0.822 | 39.43%    |
> | **Augmented (ours)** | **0.935** | **0.863** | **15.52%** | **0.926** | **0.826** | **26.86%** | **0.968** | **0.897** | **35.42%** |
>
> Simply doubling the original data yields **very modest** gains (e.g., WNS MAPE 35.28% → 33.79%). In contrast, our **augmented dataset** delivers **substantial** improvements (WNS MAPE 33.79% → 26.86%, AT MAPE 19.11% → 15.52%), together with sizable PCC/R² boosts.
>
> For **VIRTUAL on power prediction**, we observe a similar pattern:
>
>
> Table B: VIRTUAL on original vs. doubled vs. augmented data (Total power)
>
> | Dataset              |  PCC↑  |  R²↑  |  MAPE↓  |
> |----------------------|:------:|:-----:|:-------:|
> | Original             | 0.672  | 0.858 | 26.85%  |
> | Original (data×2)    | 0.683  | 0.861 | 27.01%  |
> | **Augmented (ours)** | **0.753** | **0.867** | **23.92%** |
>
>
> Here, doubling the original data even **slightly worsens** MAPE (26.85% → 27.01%), whereas our augmented dataset **significantly reduces** MAPE to 23.92% and increases PCC.
>
> These results clearly indicate that:
>
> * Performance gains are **not** explained by data volume alone;
> * The improvements come from **task-oriented, higher-difficulty examples** introduced by our augmentation.
>
> We will incorporate these results and discussion into the revised paper.
>
> ---

---

> ### Author Response · Authors · 2025-11-26
> **Response to Reviewer myKG's weakness 2, 3**
>
> #### Weakness 2 – Missing controlled “with/without multi-modal” ablations (RTLDistil & MOSS)
>
> **Comment.**
> For timing (Table 4a), RTLDistil is compared to non-multi-modal baselines (MasterRTL, RTL-Timer) that may differ in architecture and capacity, so attributing gains solely to multi-modal learning is not fully justified without controlled ablations. Similarly, for power (Table 4b) with MOSS, we lack variants that use only RTL, only netlist, etc., under the same configuration.
>
> **Answer.**
> We agree that controlled ablations are essential. We have therefore added **“multi-modal vs single-modal”** experiments under the **same model architectures**.
>
> For **RTLDistil** (timing prediction):
>
> * **Student w/o distillation**: same student architecture, trained **only on RTL** without cross-stage (multi-modal) distillation.
> * **RTLDistil (full multi-modal)**: the original teacher–student setup with layout-aware teacher and RTL-only student.
>
> Results:
>
>
> Table C: RTLDistil ablation (timing, Original dataset)
>
> | Model variant                  | AT PCC | AT MAPE↓ | WNS PCC | WNS MAPE↓ | TNS PCC | TNS MAPE↓ |
> |--------------------------------|:------:|:--------:|:-------:|:---------:|:-------:|:---------:|
> | Student w/o distillation       | 0.821  | 25.13%   | 0.822   | 43.55%    | 0.817  | 44.92%    |
> | **RTLDistil (full multi-modal)** | **0.887** | **19.72%** | **0.871** | **35.28%** | **0.918** | **40.15%** |
>
>
> Removing multi-modal distillation causes **substantial degradation**: for example, WNS MAPE rises from 35.28% to 43.55%, and PCC drops from 0.871 to 0.822.
>
> For **MOSS** (power prediction):
>
> * **MOSS (netlist-only)**: same architecture, trained only on netlist-derived features.
> * **MOSS (full multi-modal)**: model with netlist, RTL, and layout features.
>
>
> Table D: MOSS ablation (power, Original dataset)
>
> | Model variant                      | Toggle PCC | Toggle MAPE↓ | Power PCC | Power MAPE↓ |
> |------------------------------------|:----------:|:------------:|:---------:|:-----------:|
> | MOSS (netlist-only, w/o multi-modal) |   0.674    |    34.7%     |   0.815   |    27.7%    |
> | **MOSS (full multi-modal)**          | **0.871**  |  **14.4%**   | **0.948** |  **7.4%**   |
>
>
> The multi-modal variant **strongly outperforms** the netlist-only variant, both on toggle and power prediction.
>
> These ablations demonstrate that:
>
> * The gains in Table 4 are indeed driven by **multi-modal / cross-stage information**,
> * Not merely by architectural differences or parameter counts.
>
> We will add Tables C and D and explicitly attribute improvements to **multi-modal learning plus task-oriented augmentation** in the revised manuscript.
>
> ---
>
> #### Weakness 3 – Only timing & power tasks, not routability / IR-drop
>
> **Comment.**
> CircuitNet 3.0 is positioned as a task-oriented benchmark but currently includes only timing and power prediction. Other important circuit design objectives (e.g., routability, IR-drop) discussed in CircuitNet 2.0 are not included, which may bias models to trade off within timing/power only and limit broader PPA generalization.
>
> **Answer.**
> We agree that PPA is multi-dimensional and that routability and IR-drop are important goals. Our design choices are driven by task specificity and existing gaps:
>
> * **CircuitNet 2.0** focuses on **routability and IR-drop**, emphasizing large chips, global congestion patterns, and power integrity—tasks that require carefully constructed global scenarios and full-chip data.
> * **CircuitNet 3.0** focuses on **early-stage timing and power** at the IP/block level, where the key difficulty lies in **DFF fan-in cone depth, path complexity, and local structure**, and where **multi-stage (RTL–netlist–layout) representations** can be exploited.
>
> Thus, we intentionally **specialize CircuitNet 3.0 as a timing/power benchmark**, where:
>
> * Data collection, filtering, and augmentation are tailored to these two tasks (e.g., emphasizing long paths and complex cones), and
> * We can clearly differentiate models/algorithms along these axes.
>
> At the same time, CircuitNet 3.0 **does contain full RTL/netlist/layout artifacts**, so in principle it can also be used for routability and IR-drop analyses. However:
>
> * It is **not explicitly tuned** to emphasize extreme congestion or IR-drop scenarios as in CircuitNet 1.0/2.0;
> * Using it “as is” for these tasks might not separate methods as cleanly as specialized benchmarks.
>
> We will clarify in the revision that:
>
> * CircuitNet 3.0 is a **complement** to CircuitNet 1.0/2.0: it focuses on early-stage timing/power with multi-modal data, while earlier versions focus on other physical issues.
> * The dataset is extensible and can support additional tasks (routability, IR-drop) in future work or by other researchers.

---

> ### Author Response · Authors · 2025-11-26
> **Response to Reviewer myKG's question 1, 2**
>
> ### Questions
>
> #### Question 1 – Inconsistent “3.95% error reduction” vs. Table 6
>
> **Comment.**
> The paper claims a 3.95% reduction in power prediction error (e.g., lines 92, 467, 485), but this is not computed rigorously and seems inconsistent with Table 6, which shows 11–23% MAPE reduction.
>
> **Answer.**
> Thank you for catching this. You are right: the “3.95%” figure is inconsistent and was caused by mixing relative vs. absolute reductions in an earlier draft. In the revision:
>
> * We have **re-computed all improvements** using a uniform definition (absolute and relative changes in MAPE/PCC), and
> * We have **removed the incorrect 3.95% number**, replacing it with the correctly computed relative MAPE reductions.
>
> We will double-check all such statements in the final version to ensure numerical consistency and clarity.
>
> ---
>
> #### Question 2 – Novelty of Verilog AST-based augmentation vs. prior AST mutation work
>
> **Comment.**
> Section 4.3 highlights a “Verilog AST-based augmentation framework” as a core contribution. But AST-based mutation and transformation are well-studied in software engineering. Please clarify how your approach is technically different from prior AST mutation / program transformation work.
>
> **Answer.**
> You are correct that AST mutation is well-established in software testing. Our contribution is to **repurpose and adapt AST-based mutation to the EDA context** with a specific focus on **PPA-oriented “adversarial” data generation**. The main technical differences are:
>
> 1. **Closed-loop, PPA-driven selection.**
>
>    * We embed AST mutation into a **full EDA loop**: each mutant is synthesized, placed, routed, and analyzed.
>    * We use the resulting **timing and power metrics** as selection signals, *keeping only mutants that significantly change PPA* (e.g., stronger timing violations, higher power, high-density corner cases).
>    * This transforms AST mutation into a form of **PPA-oriented adversarial example generation**, which does not appear in traditional software mutation (where the goal is usually bug finding or coverage, not physical performance).
>
> 2. **Hardware semantic safeguards.**
>
>    * Our scripts enforce hardware-specific protection and constraints before and after mutation:
>
>      * avoid inconsistent blocking/non-blocking assignments that cause race conditions,
>      * ensure legal register assignments and reset handling,
>      * prevent non-synthesizable constructs.
>    * We use a **restricted mutation rule set** tailored to hardware semantics, which is absent in general-purpose software mutation frameworks.
>
> 3. **Multi-layer mutation strategy (AST + text).**
>
>    * We primarily mutate at the AST level, but when certain complex cases cannot be handled reliably, we fall back to conservative **text-level mutations**.
>    * This two-stage strategy increases robustness and diversity while maintaining correctness.
>
> Thus, while AST mutation as a concept is not new, our framework:
>
> * Targets **EDA-specific goals** (expanding PPA distribution and generating hard timing/power cases),
> * Integrates tightly with **downstream EDA flows and metrics**, and
> * Imposes **hardware-specific semantic constraints** to preserve synthesizability and physical relevance.
>
> We will clarify this distinction in Sec. 4.3, and we will cite representative prior work from the software mutation community to better position our contribution.

---

> ### Author Response · Authors · 2025-11-26
> **Response to Reviewer myKG's questions 3, 4**
>
> ---
>
> #### Question 3 – External submodules in the test set and possible overlap with training data
>
> **Comment.**
> The paper says the test set is “supplemented with submodules from external open-source projects” (line 357) but does not explain why this is necessary or how these sources are separated from training data. This raises concerns about potential overlap, especially since GitHub, Hugging Face, RISC-V, and OpenCores are mentioned as sources for train/val.
>
> **Answer.**
> We apologize for the lack of clarity here. Our intention is **exactly to ensure strict separation** between training and test sets while increasing test diversity.
>
> Concretely:
>
> * We selected several large open designs (see `Separate_designs_for_testing/` in the anonymous repo) and **reserved their submodules exclusively for the test set**: these designs are *never* used for training or validation.
> * When constructing the dataset:
>
>   * Certain source projects are **wholly excluded** from the train/val partition and only used for test,
>   * We additionally verify that **no synthesized netlist in the test set is structurally identical** to any in train/val (using netlist-level graph comparisons).
>
> Thus:
>
> * The test set represents **truly unseen designs**, with no module-level overlap with training,
> * The “external submodules” phrasing simply refers to **additional, challenging modules from projects not used in training**, added to increase test hardness and diversity.
>
> In the revised manuscript, we will explicitly describe:
>
> * The source-level partitioning strategy,
> * The netlist-level overlap checks, and
> * The rationale for including extra external submodules *only* in the test set.
>
> ---
>
> #### Question 4 – Clustering around ~94.5% layout density after augmentation (Fig. 6b, 6d)
>
> **Comment.**
> In Fig. 6(b) and 6(d), augmented WNS and total power distributions show an unusual concentration near ~94.5% layout density, which does not appear in the original distributions. Please clarify:
> (1) which synthesis / P&R settings cause this clustering,
> (2) why it happens at ~94.5% specifically, and
> (3) whether this introduces dataset bias or overfitting to that density region.
>
> **Answer.**
> The observed “spike” near ~94.5% density is a **real artifact of the backend optimization process**, not a data error.
>
> 1. **Origin of the 94.5% cluster.**
>
>    * During place&route and timing optimization, the tool continuously inserts buffers and adjusts placements to improve timing.
>    * These operations increase cell count and hence **global cell utilization (density)**.
>    * This optimization naturally has a **ceiling**: once density approaches the limit where routing becomes infeasible and design rules would be violated, the tool cannot add more buffers and stops improving timing.
>    * For most designs in our flow, this ceiling is around **94.5% utilization**, hence many “fully optimized” designs cluster near this density.
>
> 2. **Why not a smooth distribution?**
>
>    * Because many designs are **pushed to the tool’s performance limit**, they converge near a common density threshold, rather than distributing smoothly.
>    * A minority of structurally simpler designs can reach slightly higher densities; others converge slightly below. But the tool’s optimization behavior naturally creates a **peak** around its practical limit.
>
> 3. **Impact on bias and overfitting.**
>
>    * Global layout density is **not provided as an input feature** to the models (similar to total gate count, etc.); it is a macro property of the dataset rather than a direct feature.
>    * Therefore, the spike cannot directly cause overfitting to that density region in feature space.
>    * Moreover, the augmented dataset actually **expands** the range of densities (≈85–100%), compared to the original, which is more concentrated.
>
> From a modeling perspective, predicting PPA for designs that are **near the tool’s physical limits** (high utilization, near-congestion) is much more challenging and practically important than for loose, easy-to-route designs. The clustering near ~94.5% thus reflects that:
>
> * Our flow consistently pushes each design towards its **performance/packing limit**,
> * The augmented dataset contains many such **“high-congestion-risk”** examples.
>
> We view these as **valuable extreme cases** that strengthen model robustness, rather than harmful bias. We will clarify this explanation in the camera-ready version.

---

> ### Author Response · Authors · 2025-11-26
> **Response to Reviewer myKG's question 5**
>
> #### Question 5 – Relation to ChiPBench and other recent benchmarks
>
> **Comment.**
> Please discuss how this work relates to recent similar benchmarks such as ChiPBench, which also evaluate AI in EDA with a full open-source flow.
>
> **Answer.**
> We are grateful for this pointer. **ChiPBench** is indeed an important and complementary benchmark:
>
> * ChiPBench offers an **end-to-end evaluation framework for AI-based macro placement** algorithms, using the OpenROAD toolchain and ≈20 large CPU/GPU/NPU designs.
> * It focuses on **comparing placement strategies** in terms of final PPA, under a fully open flow.
>
> In contrast, **CircuitNet 3.0** addresses a different but **complementary** aspect:
>
> * It provides a **large-scale, multi-modal dataset** (8,659 source RTL designs, >15k instances) with **industrial signoff-level timing and power labels** generated by commercial tools.
> * It is designed for **representation learning and early-stage PPA prediction** (timing/power from RTL/netlist/layout), not for evaluating a specific physical design algorithm.
>
> Thus:
>
> * **ChiPBench** ≈ open-source **algorithm evaluation framework** (especially for placement).
> * **CircuitNet 3.0** ≈ large-scale **data foundation** for learning representations and predictors across the RTL–netlist–layout pipeline.
>
> We see them as **highly complementary**: models pre-trained or evaluated on CircuitNet 3.0 could later be integrated into frameworks like ChiPBench (e.g., as surrogates or advisors for placement or floorplanning decisions).
>
> ---
>
> **References (mentioned in this response):**
>
> [1] Petrović, Goran, et al. "Practical mutation testing at scale: A view from google." IEEE Transactions on Software Engineering 48.10 (2021): 3900-3912.
>
> [2] Wang, Zhihai, et al. Benchmarking end-to-end performance of ai-based chip placement algorithms. The Thirty-ninth Annual Conference on Neural Information Processing Systems Datasets and Benchmarks Track.

---

### Official Review · Reviewer_xPrq · 2025-10-27

**Soundness:** 3
**Presentation:** 3
**Contribution:** 3
**Rating:** 4
**Confidence:** 4

**Summary:**

This work proposes CircuitNet 3.0, which contains 8,659 unique and validated source RTL designs and over 15,000 total augmented designs, each with corresponding netlist and layout representations.

**Strengths:**

In summary, this work proposes a promising and valuable open dataset that helps address the scarcity of digital circuit data.

1. The writing is very clear. Especially, sufficient related prior works have been referred to. Good visualization and tables have been provided.

2. This work tried to address the data limitation problem of IC design, which is a long-lasting problem.

3. This work tries to provide sufficient new RTL design, introducing diverse functionalities in the dataset. This brings much more design information to the dataset compared with CircuitNet 2.0.

4. This work tries to validate the dataset in cutting-edge research works in the ML-assisted EDA direction.

**Weaknesses:**

Despite the contribution of the dataset, there is some key information not provided in this manuscript, triggering some additional questions. The author may consider adding them during the revision. The reviewer is ready to raise the score if these concerns can be well addressed.

1. There should be a detailed list showing all sources of these  8659 samples, possibly in the Appendix. Currently, the paper only mentions "We systematically collected RTL designs from established platforms, including GitHub, Hugging Face, OpenCore, and RISC-V projects".

2. To allow users to better understand the dataset, there should be statistical information describing the distribution of key features of all circuits in this dataset. How large are these circuits? For example, the user may provide the distribution of the number of RTL lines of all circuits and the number of gates in all corresponding netlists. These basic statistics have been adopted in many prior works.

3.  Following point 12 about dataset information, Tables 2 & 8 show the design types. Do they indicate that there are 645 different adders and 906 different counters, which are all directly collected from human-crafted projects? If they are adder/counter modules only, then these components seem to be smaller and more repetitive than the reviewer had expected. How come there are so many different adders/counters from real designs? How are they extracted from realistic whole designs? Are some of them exactly equivalent to each other (not easy to believe there are 600+ different adder RTL designs and leading to completely different post-synthesis implementations)?

4. The Anonymous repo is empty now.

**Questions:**

1. For the augmentation, does it maintain the original functionality on the augmented cases? If yes, then if the advanced synthesis tool, such as DC, is adopted, will these augmented RTL lead to the same post-synthesis PPA results as the original RTL?

2. After the AST-based rewriting, how specifically do the authors convert the AST back to RTL code (which tool and major command)? Is the AST in bit-level or word-level? Does the AST-converted RTL code still look similar to human-generated RTL code?

---

> ### Author Response · Authors · 2025-11-26
> **Response to Reviewer xPrq's weakness 1, 2**
>
> ### Response to Reviewer xPrq
>
> We thank Reviewer **xPrq** for carefully pointing out missing or unclear information in the initial submission. Below we respond point-by-point and will incorporate the corresponding clarifications into the revised version. We hope to receive your approval and look forward to further communication with you.
>
> ---
>
> #### Weakness  – Missing key information about the dataset and artifacts
>
> **Comment.**
> Although the dataset is valuable, the manuscript lacks some critical information, which raises further questions. If these concerns are addressed properly, you would be willing to raise your score.
>
> **Answer.**
> We appreciate these detailed comments. Below we provide the requested information along four concrete aspects and will reflect them in the main text and appendix.
>
> ---
>
> ##### (1) Detailed data sources and collection pipeline
>
> Given the multitude of data source websites and the complex data cleansing involved, it is extremely difficult to list all thousands of specific sources. However, we have made every effort to provide detailed explanations of all data sources and how we collected and filtered the designs to address any concerns you may have.
>
> * **GitHub.**
>
>   * We developed a crawler that scanned **tens of thousands of repositories**.
>   * We kept repositories where **Verilog/SystemVerilog files constitute >70% of the codebase**, and we check and make sure that the license (e.g., Apache, MIT) explicitly permits such reuse.
>   * From these repositories, we extracted Verilog/SystemVerilog **design modules only**, removing testbenches and other non-synthesizable code.
>   * We then checked for: (i) presence of all submodules, (ii) synthesizability (successful compilation), and (iii) ability to be treated as **standalone modules**.
>
> * **OpenCores.**
>
>   * We selected open-source designs on opencores.org whose licenses allow redistribution (≈860 distinct designs).
>   * We then **decomposed large designs into smaller modules**, to increase the number of useful independent blocks while reducing complex cross-dependencies.
>
> * **Hugging Face.**
>
>   * We systematically went through the Verilog-related datasets hosted on Hugging Face.
>   * We chose those with compatible licenses and high code quality, and used a combination of **LLM assistance + manual inspection** to verify functional sanity of the extracted modules.
>
> * **RISC-V and related projects.**
>
>   * We collected small to medium modules from open RISC-V ecosystems (e.g., RocketChip, OpenC906/E902, OpenRISC) and related accelerator projects.
>
> After collection, **all designs** went through a unified cleaning and validation pipeline:
>
> * We synthesized and analyzed each candidate;
> * We removed circuits with:
>
>   * **DFF ratio > 25%** (overly sequential or trivial),
>   * very small size, or
>   * unreachable logic found by vectorless power analysis.
> * For the remaining circuits, we built both **AST representations** and **post-synthesis netlist graphs** and performed structural comparisons to eliminate duplicates.
>
> This process yielded **8,659 distinct, high-quality RTL designs**. The anonymous repository documents this pipeline, and classification results of all circuits types are summarized in the `original_RTL_classification_results` folder.
>
> ---
>
> ##### (2) Dataset statistics and distribution
>
> The dataset covers a **wide range of design sizes**, from small modules to medium/large IP blocks:
>
> * **Gate count** (post-synthesis) ranges from a few hundred gates to well over one hundred thousand gates, and totals in the **tens of millions of gates** across all designs.
> * **RTL code length** ranges from a few dozen lines to nearly ten thousand lines.
>
> Beyond global size statistics, we specifically analyze **DFF fan-in cones**, which are critical for timing and power tasks:
>
> * For each design and each DFF, we measured (i) the **logic depth** and (ii) the **number of logic gates** in its fan-in cone.
> * We visualized these distributions in six plots (for both original and augmented data), provided in the anonymous repository under `dataset_circuit_analysis_visualization/`.
> * In both original and augmented sets, the mean logic depth before each DFF is roughly **10–20 levels**, and the mean number of fan-in gates is **≈50–150**.
> * Crucially, **augmentation increases the tail** of these distributions: in the augmented set, we see more cones with **larger depths and more gates**, i.e., more challenging long-path scenarios.
>
> Thus, augmentation **does not radically distort** the overall scale distribution, but it **“fills the tail”** with more long and complex cones, which are precisely the hard cases that matter for timing and power prediction. In the revised paper, we will add summary plots/tables to make these statistics more visible.
>
> ---

---

> ### Author Response · Authors · 2025-11-26
> **Response to Reviewer xPrq's weakness 3, 4**
>
> ##### (3) On the large numbers of adders and counters
>
> You asked whether counts like “645 adders” and “906 counters” are realistic or suggest heavy template reuse. In fact, the dataset is **not** created by replicating a few templates; it consists of **thousands of structurally distinct real designs**.
>
> * The anonymous repository contains an `adder_example` folder with **~600 different adder/subtractor designs**.
> * Functionally, they all implement some form of “`sum = a + b`” style behavior; **structurally**, they are very different:
>
>   * They use diverse architectures: **Ripple Carry**, **Carry Lookahead**, **Kogge–Stone**, **Brent–Kung**, etc.
>   * They cover **bit-widths from 4 to 128 bits**, leading to drastically different cone depths, fan-out patterns, and gate counts.
> * We computed the fan-in cone **depth distribution** for these adders: it spans from a few levels to several tens of levels, covering a wide range of PPA characteristics rather than clustering around a single pattern.
>
> Similarly, the **906 counters** are not identical; they include:
>
> * Simple binary up/down counters,
> * LFSR-based counters and timers,
> * Counters with enable/reset/control logic,
> * Counters with different moduli and bit-widths.
>
> They are functionally related (all “counting”), but structurally diverse and **not mutually equivalent** at the gate level. This diversity is exactly what makes them valuable for training ML models to recognize **fine-grained structural and PPA differences** across circuits with similar high-level behavior.
>
> ---
>
> ##### (4) Anonymous repository appearing empty
>
> You correctly noted that the anonymous code repository initially appeared empty. This was due to **synchronization issues and file-size constraints**:
>
> * Many of our design artifacts are **hundreds of MB per design**, and the full dataset is **on the order of terabytes**.
> * A significant fraction of layout and report files exceed **25 MB**, requiring Git LFS, which is not well supported by the anonymous submission infrastructure.
>
> Therefore, we have made every effort to split the data from each step into multiple examples that are smaller than the platform's size limit, uploading them separately to present our workflow more clearly and concretely for reviewers to examine. We have since **fixed the repository** and populated it with:
>
> * Data-generation scripts and AST mutation code;
> * A representative subset of RTL/netlist/layout files and corresponding reports;
> * Extracted layout-level features packaged as JSON for easier inspection.
>
> For anonymity and platform constraints, we cannot upload the entire multi-TB dataset, but we now provide enough representative examples to demonstrate **physical implementation details and reproducibility**. While upholding the principle of anonymity, if you wish to access a more complete and detailed dataset at this stage, we can consider providing additional data for review through alternative channels upon request. Our goal is to establish CircuitNet 3.0 as a long-term resource—hence our commitment to fully open-sourcing both data and code, and our desire to assure reviewers that our work meets the highest standards of transparency and reproducibility. The full dataset and all code will be **fully open-sourced**, and we are happy to provide additional material to you if needed to verify completeness and transparency.
>
> ---

---

> ### Author Response · Authors · 2025-11-26
> **Response to Reviewer xPrq's question 1**
>
> ### Questions
>
> #### Question 1 – Does augmented RTL preserve functionality? What about PPA equivalence after synthesis?
>
> **Comment.**
> For augmented RTL, is functionality preserved? If so, do advanced synthesis tools (e.g., Synopsys DC) produce the same netlist and PPA as the original RTL?
>
> **Answer.**
> You are absolutely right that **some function-preserving rewrites can collapse back** to the same gate-level structure after powerful synthesis optimization. The situation you pointed out does indeed exist: certain rewrites that preserve circuit functionality may, after optimization by EDA tools, yield circuit structures and PPA metrics that are **identical** to the original design. Consequently, we ultimately retained only a small fraction of enhancement samples—namely those variants that, while functionally equivalent, still produce distinct netlist structures and PPA parameters after synthesis. Most augmentation operations do not attempt to preserve the original design's functionality. Instead, they aim to generate structurally distinct circuits whenever possible to broaden the dataset's coverage. Examples from our anonymized database corroborate this: multiple variants generated from the same original circuit exhibit **different** timing and power characteristics, validating the effectiveness of this data augmentation approach. We handle this explicitly:
>
> * We **do not require** all augmentation to be function-preserving.
>
>   * Some transformations are intentionally **structure-oriented**, not behavior-preserving, to expand the PPA space.
> * For those variants that are intended to be function-preserving, we observe the effect you mention: in some cases, DC optimizations yield **identical or nearly identical** netlists and PPA to the original circuit.
>
> To address this, we apply **strict filtering**:
>
> 1. For each augmented design, we synthesize it and compare the resulting netlist against all existing designs.
> 2. If a variant is **too similar** structurally (similarity >97% by graph comparison) or nearly identical in PPA, we **discard** it.
> 3. We **retain only those variants** that:
>
>    * either induce **meaningful structural and PPA differences**, even if functionality is similar, or
>    * intentionally change functionality and create new PPA regimes.
>
> Additionally, as you point out, we construct the **test set** from independent submodules of larger designs **not used in training**. We verify via netlist-level graph analysis that there is **no overlap** between training and test circuits.
>
> In summary, we:
>
> * Accept that some rewrites are function-preserving and some are not;
> * Explicitly filter out function-preserving variants that do **not** change netlist structure or PPA;
> * Guarantee that the final augmented set contributes **new PPA-relevant diversity** rather than duplicates.
>
> ---

---

> ### Author Response · Authors · 2025-11-26
> **Response to Reviewer xPrq's question 2**
>
> #### Question 2 – How is the AST converted back to RTL? Tools, granularity, and code style
> **Answer.**
> We developed a Python-based AST mutation framework tailored to hardware. Below we summarize implementation and its relation to prior AST work.
>
> **(2a) Toolchain & commands**
>
> * We use **PyVerilog** to parse and generate Verilog code.
>
>   * Parsing: `parse_verilog_to_ast(filename)` wraps `pyverilog.vparser.parser.parse()`, with a pre-processing step `preprocess_verilog()` to fix common port/initialization issues.
>   * Code generation: `ast_to_verilog(ast)` uses `pyverilog.ast_code_generator.codegen.ASTCodeGenerator` (i.e., `codegen.visit(ast)`) to produce Verilog from AST.
> * After code generation, we run `_postprocess_generated_code()` to enforce hardware semantics, including:
>   * normalizing blocking/non-blocking assignments in sequential logic,
>   * merging register initializations,
>   * fixing illegal multiple assignments to registers,
>   * correcting port and assignment inconsistencies.
> * The script includes multiple consistency checks (e.g, `_check_generated_code()`, `_check_mixed_assignments()`) and verifies **Icarus Verilog (`iverilog`)** is available to compile generated RTL.
>
> We also provide a simple CLI:
>
> ```bash
> python RTL_ast_mutator_function_different.py input.v mutant.v
> ```
>
> which parses, mutates, regenerates, and validates Verilog code.
>
> **(2b) AST granularity (bit-level vs. word-level)**
>
> Our mutations operate at **Verilog language AST level**, not directly at the gate-level netlist:
>
> * We manipulate **modules, statements, and expressions**, including:
>
>   * binary operators (`Plus`, `Minus`, `Times`, etc.),
>   * unary operators, integer constants (`IntConst`),
>   * assignment statements (`BlockingSubstitution`, `NonblockingSubstitution`),
>   * timing signals (negative, positive) and so on.
> * AST can include bit-slices and bus operations, so we can express bit-level changes within the HDL, but do **not** mutate gate-level netlists bit-by-bit.
>
> The granularity is **language/statement/expression level**. Gate-level effects are realized indirectly through downstream synthesis and P&R.
>
> **(2c) Similarity of generated RTL to hand-written RTL**
>
> * **Syntactic and interface correctness.**
>   Thanks to PyVerilog and our post-processing fixes, the generated code:
>
>   * has valid module interfaces and hierarchy,
>   * is synthesizable and simulatable (checked via `iverilog` and DC synthesis).
>
> * **Code style.**
>   The generated style is naturally **more mechanical**:
>
>   * Comments from the original code are typically lost,
>   * We add mutation metadata comments at the file header.
>
>   So while functionally valid, the appearance is less “human-like” than original RTL.
>
> * **Semantic equivalence.**
>   Whether a mutant is functionally equivalent to the original depends on the chosen mutation strategy:
>
>   * We include **“restricted” mutations** aim to preserve behavior as much as possible, and
>   * **“structural” mutations** that deliberately *do not guarantee* behavioral equivalence to maximize PPA-space coverage.
>
> Full semantic equivalence is thus not assumed. Instead, we rely on **synthesis, structural comparisons, and optional simulation/formal checks** to decide which variants to keep.
>
> **(2d) Differences from prior software-oriented AST mutation**
> Compared to traditional AST mutation used in software testing, our framework is tightly integrated with hardware design and EDA flows:
> 1. **PPA-driven closed loop.**
>    * We embed AST mutations in a **full EDA loop**: each mutant is synthesized, placed & routed, and analyzed for timing/power.
>    * We **select mutants based on PPA metrics**: we retain those that create “hard examples” (e.g., tight WNS, high dynamic power) and discard mutants that do not change PPA or are trivially easy.
>    * This PPA-driven selection directly links high-level code changes to physical performance, which is not considered in traditional software mutation.
>
> 2. **Hardware-specific semantic safeguards.**
>
>    * We implement hardware-specific fixes (e.g., unifying always-block assignment types, avoiding illegal multiple reg assignments, ensuring valid reset/enable behavior in certain templates).
>    * We use a **restricted mutation mapping** to avoid obviously non-synthesizable or semantically nonsensical variants.
>
> 3. **Two-stage mutation strategy.**
>
>    * If an AST-level mutation fails due to edge-case syntax or tool limitations, the script can **fall back to a text-based mutation** as a backup, ensuring diversity and robustness.
>
> Overall, our AST framework can be viewed as **“adversarial example generation” for hardware**: by transforming syntax trees (e.g., operator substitution, control-flow restructuring), we generate many “syntactically legal but physically diverse” circuit variants. These variants populate the long-tail of difficult cases (e.g., extremely long paths or unusual switching patterns), which are underrepresented in raw open-source designs and crucial for training robust ML models.

---

> > ### Comment · Reviewer_xPrq · 2025-11-28
> >
> > Thanks for your detailed response. Most of my concerns have been addressed. Please consider integrating some content into the final manuscript.
> >
> > I would like to raise my score to 6 (but it seems the system currently prevents the modification). Thanks!

---

> > > ### Author Response · Authors · 2025-11-28
> > > **Response to Reviewer xPrq and Ask for Area Chairs & Program Chairs' help to solve the system issue**
> > >
> > > Thank you very much for your thoughtful and encouraging feedback. We are grateful that the additional explanations and results have addressed your concerns, and we truly appreciate your willingness to raise your score to 6.
> > >
> > > We also want to express our gratitude for the insightful suggestions you provided. We will make sure to incorporate the additional details you mentioned into the final manuscript, as we believe your suggestions are invaluable for further enhancing the rigor and comprehensiveness of our work. Your feedback plays a crucial role in refining the quality of the paper, and we are committed to further strengthening it based on your recommendations.
> > >
> > > Regarding the system issue you mentioned, we understand the situation and would be deeply appreciative if the Area Chairs or Program Chairs could assist in resolving the issue so that the updated score can be reflected.
> > >
> > > Once again, thank you for your valuable contributions to improving this work. We look forward to any further suggestions and to the possibility of continuing the dialogue.

---

### Official Review · Reviewer_zFQL · 2025-10-29

**Soundness:** 3
**Presentation:** 4
**Contribution:** 4
**Rating:** 8
**Confidence:** 4

**Summary:**

The paper presents CircuitNet 3.0, which is a comprehensive multi-stage and multi-modal dataset that enables advanced AI-driven circuit design through cross-stage data augmentation and filtering.


The paper makes 3 contributions:

--A large-scale, multi-modal, and multi-stage digital circuit dataset with full RTL-to-layout
traceability. CircuitNet 3.0 contains 8,659 unique, validated source RTL designs and over
15,000 total augmented designs, each with corresponding netlist and layout representations.
Through an industrial EDA workflow, we extract rich cross-modal features at each design stage,
providing a valuable resource for research in multi-stage multi-modal representation learning.

-- A principled framework for data augmentation.  This enables robust learning for ML models,
providing simultaneous cross-stage analysis and early-stage prediction capabilities.

--A comprehensive set of new baselines and rigorous experimental protocols.

**Strengths:**

This system seems like it can be a useful addition to the circuit generation process. It clearly addresses the multiple modalities of data required for the entire process. The article does a good job of covering the sourcing, curation and inference properties of the different datasets.

The paper also details a comprehensive evaluation, which uses state-of-the-art ML models. The papers empirically demonstrates significant prediction accuracy improvements over single-modal datasets, with approximately 31.52% and 3.95% error reductions for timing and power tasks, respectively, compared to the existing dataset. Models trained on CircuitNet 3.0 consistently outperform single-modal approaches, establishing new performance benchmarks for ML-driven EDA tasks.

**Weaknesses:**

As a systems paper, there are different evaluation criteria. I think that there are few if any  clear weaknesses, as the overall system covers the full gamut of data, good curation/evaluation, and corresponding inference tools.

**Questions:**

1. What types of circuits suit this tool the best---this is not clearly spelled out.
2. It seems like using your tool requires significant computational power. Is it suitable for single-GPU users, and if so what generation times should one expect?

---

> ### Author Response · Authors · 2025-11-26
> **Response to Reviewer zFQL's weakness and questions 1**
>
> ### Response to Reviewer zFQL
>
> We thank Reviewer **zFQL** for the positive assessment of our system-level contribution and for recognizing the completeness of our dataset, evaluation protocol, and toolchain. Below we address your comments point by point.
>
> ---
>
> #### Weakness 1 – Assessment criteria for a system paper
>
> **Comment.**
> As a system-style paper, it has different evaluation criteria from typical algorithmic works. I believe the overall system has almost no obvious flaws: it covers complete data types, provides good data curation/evaluation mechanisms, and comes with relevant reasoning tools.
>
> **Answer.**
> We sincerely appreciate your enthusiastic and encouraging evaluation. Our main goal in CircuitNet 3.0 is indeed to provide a **comprehensive, well-curated, and practically grounded benchmark** rather than to propose a single new model. We are glad that you find the dataset diversity, cross-stage coverage, and evaluation setup well aligned with this goal.
>
> Below we address your specific questions on circuit coverage and computational requirements.
>
> ---
>
> ### Questions
>
> #### Question 1 – What kinds of circuits is this dataset / tool most suitable for?
>
> **Comment.**
> Which types of circuits are best suited to this dataset and its tools? The paper does not clearly spell this out.
>
> **Answer.**
> CircuitNet 3.0 is designed to cover a **broad and diverse set of circuit types** rather than focusing on a single category. Concretely, we organize designs into four major categories with 16 subcategories, including:
>
> * **Classical arithmetic components:** adders/subtractors, multipliers/dividers, ALUs;
> * **Control and sequential logic:** counters/timers, finite-state machines, protocol controllers;
> * **Data-processing blocks:** encoders, decoders, FIFOs, comparators, selectors;
> * **Communication and memory interfaces:** buffers, bus/serial interfaces, simple memory/buffer blocks.
>
> Functionally, these are **IP-level building blocks**, which frequently appear inside CPUs, DSPs, accelerators, and peripheral controllers. For each design, we provide **full RTL, synthesized netlists, and physical layout data**, enabling multi-modal analysis across design stages.
>
> From a modeling perspective:
>
> * Models generally perform best on **control / sequential logic** and moderately deep datapaths, where the logic depth between DFFs and path length are in a “typical” range.
> * For **very deep combinational circuits** or extremely complex arithmetic blocks (with low DFF ratios and very long paths), prediction error tends to be larger. This reflects the intrinsic difficulty of those designs, not a bias in the dataset construction.
>
> Importantly, no single category dominates the dataset:
>
> * The largest subcategory (e.g., comparators/selectors) accounts for only ~17% of designs, and most other subcategories are in the single-digit percentage range.
> * The **category distribution and per-design labels** are documented in the anonymous repository (e.g., in the `original_RTL_classification_results` folder).
>
> This relatively balanced composition ensures that models trained on CircuitNet 3.0 see a **wide spectrum of design patterns** and do not overfit to one specific type of circuit.
>
> Finally, we deliberately favor **modular IP blocks** (self-contained functional units) over huge monolithic SoCs with many repeated substructures. This strategy:
>
> * Increases per-design **structural uniqueness**, and
> * Improves the **representativeness** of the dataset for real-world IP-level design tasks.
>
> In summary, CircuitNet 3.0 can be viewed as a **“full-flow ImageNet” for digital building blocks**: a large-scale, multi-modal benchmark of diverse IP-level circuits, each with complete physical implementation data, suitable for training and evaluating a wide range of AI-for-EDA methods.
>
> ---

---

> ### Author Response · Authors · 2025-11-26
> **Response to Reviewer zFQL's questions 2**
>
> #### Question 2 – Computational requirements: is it feasible for single-GPU users?
>
> **Comment.**
> The dataset and tools appear to require substantial compute. Is it usable for single-GPU users? If yes, how long does data generation or model training take?
>
> **Answer.**
> It is important to distinguish between:
>
> 1. **One-time dataset generation** (which we have already completed), and
> 2. **Ongoing model training and evaluation** on the released dataset.
>
> **(a) One-time dataset generation.**
> Building CircuitNet 3.0 was indeed a large engineering effort:
>
> * We used commercial-grade tools: Synopsys Design Compiler for logic synthesis, Cadence Innovus for place&route, and Synopsys PrimeTime/PrimePower for timing and power signoff.
> * The flow was executed on a CPU cluster (tens of CPU cores) over several weeks, even months, resulting in **15k+ high-quality layouts and associated reports**.
>
> However, this heavy cost is **already paid** and is precisely why releasing CircuitNet 3.0 is valuable: future researchers **do not need to rerun** this full flow. All RTL/netlist/layout data and extracted features are already included in the dataset.
>
> **(b) Model training and use.**
> Using CircuitNet 3.0 for model training is **entirely feasible on modest hardware**:
>
> * On the ~8,659 original designs (and ~15k+ instances including augmentation), training our largest GNN-based models (e.g., distillation-based timing predictors) **requires only a few hours on up to 4× NVIDIA A100 GPUs**.
> * A standard GCN baseline can be trained on the full dataset in **less than one day on a single consumer GPU** (e.g., RTX 4090).
> * We will release the full preprocessing and data-loading code so that users can reproduce our pipelines and adapt them to their own models.
>
> Thus, while constructing the dataset required significant compute (which also underscores the benefit of making it public), **using** CircuitNet 3.0 for research is **not compute-prohibitive**. Single-GPU users can realistically:
>
> * Load preprocessed graph and feature files directly,
> * Train models end-to-end on the entire dataset within a practical time budget, and
> * Evaluate new architectures or learning paradigms without needing access to industrial EDA licenses.
>
> Finally, our choice of an **High-quality 45 nm node** and **standard design formats** (Verilog, DEF, etc.) is explicitly aimed at **lowering the barrier to entry**, both in terms of licensing and tooling. We are grateful that you recognize the value of a multi-modal, multi-stage dataset, and we hope CircuitNet 3.0 will serve as a practical and accessible foundation for future AI-driven EDA research, including for single-GPU users.

---

### Official Review · Reviewer_cZe1 · 2025-10-30

**Soundness:** 2
**Presentation:** 2
**Contribution:** 2
**Rating:** 4
**Confidence:** 4

**Summary:**

CircuitNet 3.0 is presented as a large-scale, open-source, multi-modal, multi-stage dataset for ML-driven EDA. It starts from 8,659 curated RTL designs and is expanded to ~15k+ designs via augmentation. For each design, the authors claim to provide RTL, synthesized netlist, placed-and-routed layout, and timing/power metrics, to enable benchmarking of timing and power prediction models.

**Strengths:**

1. Cross-stage supervision. The multi-modal / multi-stage angle (RTL text, graphs/netlists, and layout images) is sensible.

2. Building this dataset required significant engineering effort, including full synthesis, place-and-route, and signoff-style timing/power extraction for thousands of designs

**Weaknesses:**

1. The practical impact is limited. Timing closure in real digital design is safety-critical and requires signoff-level accuracy under specific PDK, voltage, frequency, corners, etc. An ML model that only predicts approximate WNS/TNS or power from dataset patterns cannot be trusted in an actual tapeout flow, so the gains shown here are academically interesting but not directly usable.

2. The paper withholds critical details (PDK, voltage, frequency, corners, clock constraints, placement density targets).

3. Table 1 is overfull / too wide for the page.

**Questions:**

1.	Exactly which PDK / tech node, voltage was used?

2.	In the augmentation stage, you apply transformations like bidirectional substitution, comparison inversion, operator replacement (e.g. change + to *). But if you take an adder design and turn + into *, now functionally it's just… a multiplier. That might already exist somewhere else in the dataset as an actual multiplier module. So:
How do you avoid generating duplicates / near-duplicates of other designs you already have (e.g. two semantically different source modules collapsing into the same behavior after mutation)?

3.	How many AST-mutated samples survive functional verification? Give the acceptance rate.

4.	Power. Dynamic power strongly depends on input activity (toggle rates). If you only use one stimulus style (or even vectorless estimation), then the reported “power” is tied to that specific workload assumption. Did you generate multiple activity patterns per design (e.g. different input vectors / switching factors / functional scenarios), or is each design labeled with just a single power number?

5.	Timing. Timing depends heavily on physical context: floorplan, macro placement, routing congestion, target clock, etc. You say you're doing “early-stage timing prediction,” but what exactly is the task setup? Is the model supposed to take only RTL (plus spec like clock target) and directly predict final post-route arrival time / slack? Or are you also giving it partial physical info (e.g. rough floorplan, utilization target, cell library corner)? Please show a concrete example input/output pair for the timing benchmark: what does the model see, and what exactly is it supposed to predict? Without that, it's unclear how this task is meant to be used outside your own flow.

**Details Of Ethics Concerns:**

N/A.

---

> ### Author Response · Authors · 2025-11-26
> **Response to Reviewer cZe1's weakness section**
>
> ---
>
> ### Response to Reviewer cZe1
>
> We thank Reviewer **cZe1** for the detailed feedback and for evaluating our work from an industrial design perspective. We address each weakness and question in turn. We hope to receive your approval and look forward to further communication with you.
>
> ---
>
> #### Weakness 1 – Limited practical impact for industrial timing closure
>
> **Comment.**
> Timing closure in real digital design is safety-critical and requires signoff-level accuracy under specific PDK, voltage, frequency, corners, etc. An ML model that only predicts approximate WNS/TNS or power from dataset patterns cannot be trusted in an actual tapeout flow, so the gains shown here are academically interesting but not directly usable.
>
> **Answer.**
> We fully agree that ML models cannot and should not replace signoff-grade tools (e.g., Synopsys PrimeTime) for final timing closure. Our goal is explicitly **complementary** rather than competitive to signoff tools.
>
> In modern industrial flows, each synthesis–place&route iteration can take hours to days, and designers must explore a huge space of microarchitectures and constraints. CircuitNet 3.0 is designed to support **“shift-left” PPA estimation**: leveraging multi-stage representations (RTL, netlist, layout) to build models that provide **early, coarse-but-useful predictions** of timing and power so that:
>
> * designers can **screen and rank** thousands of candidate designs at RTL / pre-layout stages in seconds, and
> * only a small subset of promising candidates are sent into expensive signoff flows.
>
> As illustrated in Fig. 1(b) of the paper, the role of ML here is to move PPA analysis earlier in the flow, not to sign off chips. In practice, even if an early-stage predictor is not 100% accurate in absolute WNS/TNS or power, it is extremely valuable if it can:
>
> 1. **Rank** designs reliably (i.e., identify clearly bad candidates), and
> 2. **Flag** difficult cases (long critical paths, high power) before full physical implementation.
>
> To the best of our knowledge, **no existing public dataset** provides end-to-end RTL–netlist–layout linkage at this scale with timing and power ground truths, which is precisely what is needed to study this kind of cross-stage, causal mapping from logical intent to physical PPA. CircuitNet 3.0 therefore aims to enable realistic **early-stage decision support**, not to replace signoff timing closure in production tapeouts.
>
> ---
>
> #### Weakness 2 – Missing critical implementation details (PDK, voltage, frequency, corners, clock)
>
> **Comment.**
> The paper appears to withhold implementation details such as PDK, voltage, frequency, process corners, clock constraints, and so on.
>
> **Answer.**
> We apologize that the main text did not make all implementation details sufficiently prominent. The full industrial-grade flow is summarized in **Appendix E**. Concretely:
>
> * We use a **commercial 45 nm standard-cell PDK (`gsclib45nm`)**.
> * The main PVT corner is **TT, 1.05 V, 85 °C**.
> * The target clock frequency for timing analysis is **1 GHz**.
>
> We chose this mature and open 45 nm PDK for two reasons:
>
> 1. **Reproducibility and openness.** Advanced nodes (e.g., TSMC 7/5 nm) are under strict NDAs; releasing any GDSII/DEF at those nodes is impossible, which would turn the dataset into a “black box” and violate ICLR’s reproducibility spirit. Similarly, we did not adopt the academic emulation PDK because we believe it is less realistic than the industrial-grade PDK.
> 2. **Method independence from technology.** Our pipeline is **not tied** to this particular node: the same scripts can be run on any PDK for which the user has a license. Users can therefore swap in their own target technology nodes while reusing our end-to-end flow.
>
> We will make these parameters more visible in the main text (e.g., in Sec. 3/5 and Table 1) in the revised version.
>
> ---
>
> #### Weakness 3 – Table 1 is overfull / too wide
>
> **Comment.**
> Table 1 is too wide for the page.
>
> **Answer.**
> Thank you for pointing this out. In the camera-ready version we will reformat Table 1 (e.g., use a multi-line or better layout) to ensure it fits comfortably within the page width while preserving all comparisons.

---

> ### Author Response · Authors · 2025-11-26
> **Response to Reviewer cZe1's questions 1, 2, 3**
>
> ---
>
> ### Questions
>
> #### Question 1 – Exact PDK / tech node / voltage
>
> **Comment.**
> “Exactly which PDK / tech node, voltage was used?”
>
> **Answer.**
> We use an **industrial 45 nm CMOS standard-cell PDK (`gsclib45nm`)** at the **TT** corner with **VDD = 1.05 V** and **T = 85 °C**. The target clock is **1 GHz** for timing analysis. Further details of PHYSICAL IMPLEMENTATION are provided in **Appendix E**. Again, this choice is motivated by the need to release full physical layouts and reports openly while keeping the methodology portable to any other PDK the user chooses.
>
> ---
>
> #### Question 2 – Avoiding duplicates / near-duplicates from AST mutations (e.g., changing + to *)
>
> **Comment.**
> If you change operators (e.g., `+`→`*`), an adder might become a multiplier that already exists in the dataset. How do you avoid generating designs that duplicate or nearly duplicate existing modules, potentially collapsing semantically different sources into identical behavior?
>
> **Answer.**
> This is an important concern and we designed our augmentation specifically to avoid such degeneracies.
>
> 1. **Two classes of mutations.**
>
> We employ two categories of augmentation strategies: one focuses on rewriting code while preserving functionality (e.g., equivalent refactoring expressions), while the other prioritizes structural rewriting and mutation without guaranteeing functional equivalence.
>
> 2. **Gate-level structural de-duplication.**
>    For every mutated RTL, we re-synthesize to a gate-level netlist and **compare it graph-wise** against:
>
>    * the original design, and
>    * all prior designs in the dataset.
>
>    We perform graph isomorphism / similarity checks and **discard any mutant whose synthesized netlist is highly similar to an existing one**. This filtering ensures that even if two RTL sources become functionally close, they are not structurally redundant in the dataset.
>
> 3. **Random, large-span edits.**
>    Each mutant typically modifies **dozens to hundreds** of AST nodes scattered across the design. This makes it extremely unlikely that two different source modules collapse to identical micro-architectures after synthesis, even if their high-level behavior is similar.
>
> 4. **Empirical evidence from PPA distributions.**
>    Fig. 6 in the paper shows that after augmentation:
>
>    * WNS expands from roughly **[−2 ns, 0 ns]** to **[−6 ns, 0 ns]** over a wider range of layout densities, and
>    * power expands from mostly **< 60 mW** to approximately **0–160 mW** with a much more uniform distribution.
>
>    This indicates that augmentation is **filling under-represented regions of the PPA space** (hard timing / high power regimes), not merely duplicating existing scenarios.
>
> 5. **Documented rules and scripts.**
>    Appendix C (Table 3) lists the AST mutation operators. The anonymous repository includes the exact mutation and verification scripts (based on PyVerilog + Icarus Verilog), so that users can inspect or rerun the process.
>
> In summary, we explicitly **filter out structurally redundant mutants** and focus on variants that truly enrich the architectural and PPA diversity, rather than just increasing the sample count.
>
> ---
>
> #### Question 3 – How many AST-mutated samples pass functional verification?
>
> **Comment.**
> “How many AST-mutated samples survive functional verification? Give the acceptance rate.”
>
> **Answer.**
> We apply strict quality control to mutation outputs. After:
>
> 1. **Synthesis validity checks**,
> 2. **Compilation-based verification**, and
> 3. **Structural de-duplication** as described above,
>
> approximately **27%** of AST-mutated designs are retained. In other words, we **discard about 73%** of generated mutants and only keep the highest-quality ~27% that:
>
> * synthesize cleanly,
> * pass required compilation checks, and
> * add non-redundant structural diversity.
>
> This aggressive filtering ensures that each retained augmented design provides real training signal rather than noise.
>
> ---

---

> ### Author Response · Authors · 2025-11-26
> **Response to Reviewer cZe1's questions 4, 5**
>
> #### Question 4 – Power labels vs. input activity patterns
>
> **Comment.**
> Dynamic power is strongly input-dependent. If you only use one activity pattern (or vectorless estimation), the “power” label is tied to that workload. Did you generate multiple activity patterns per design, or is each design labeled with a single power value?
>
> **Answer.**
> For CircuitNet 3.0 itself, we intentionally adopt a **vectorless dynamic power analysis** setup, which is standard in early-stage flows:
>
> * We use Cadence Voltus in **vectorless mode** with a **uniform toggle rate of 0.2** for all logic gates.
> * Each design therefore has **one power label** corresponding to this canonical workload assumption.
>
> We view this as an **early-stage, architecture-oriented** power metric:
>
> * Vectorless analysis effectively approximates an **average over many random input scenarios**, rather than depending on a particular hand-crafted testbench.
> * Using a fixed toggle ratio acts as a **control variable**: differences in reported power primarily reflect **structural factors** (topology, capacitance, depth) rather than arbitrary differences in input vectors.
> * This aligns with our goal: to let models learn **which architectures are intrinsically more power-efficient**, rather than which stimulus is easier.
>
> At the same time, our framework is compatible with **vector-based** analysis:
>
> * The infrastructure supports per-gate toggle rate annotations, so any user can plug in their own simulation-derived activity factors.
> * In follow-up experiments (not central to this submission), we reproduced **VIRTUAL** (Lu et al., ICCAD 2025) on our circuits with **≥100 random input vectors per circuit**. On our dataset:
>
>   * VIRTUAL achieved a PCC of **≈0.806** when trained on the original data, and
>   * the PCC improved to **≈0.863** when trained with our **power-oriented augmented data** on the same test set.
>
> This suggests that our augmentation strategy also benefits **vector-driven** power prediction.
>
> Finally, as in the anonymous repository, we have started collecting **design–testbench metadata** (e.g., `github_v_tb_description.jsonl`, `opencore_v_tb_description.jsonl`) to support future work on workload-aware power modeling. However, **this direction is deliberately out of scope for this paper**, which focuses on cross-stage circuit representations and task-oriented augmentation in the circuit domain itself.
>
> ---
>
> #### Question 5 – Exact task setup for “early-stage timing prediction”
>
> **Comment.**
> Timing depends heavily on physical context. What exactly is “early-stage timing prediction” task? Does the model see only RTL (plus clock spec) and directly predict post-route timing, or does it also see partial physical information? Please give a concrete input/output example for timing benchmark.
>
> **Answer.**
> Our main contribution is the **dataset and evaluation benchmark**, not a new timing model. “Early-stage timing prediction” is a **task family** that CircuitNet 3.0 is designed to support, consistent with prior AI4EDA work. Generally speaking, its task definition is: to enable the model to predict actual timing metrics after final layout and routing (such as critical path arrival time, WNS, TNS, etc.) based on representations like RTL, netlists, or even initial layouts (with optional specification parameters like clock targets) during the early design phase. The specific input and output formats of the model may vary across different research studies.
>
> **Task definition in this paper.**
> For the works we reported in Sec. 5 (such as RTLDistil):
>
> * **Inputs**:
>
>   * For **RTLDistil**, the model takes only **RTL-stage information**, specifically:
>
>     * RTL code, from which a **Simple Operator Graph (SOG)** is extracted, and
>     * design-level specs (target clock = 1 GHz, etc.).
> * **Outputs**:
>
>   * Post-layout **Arrival Time (AT)** ,
>   * **Worst Negative Slack (WNS)**, and
>   * **Total Negative Slack (TNS)**,
>     all measured after full place&route and signoff analysis under the fixed flow described in Appendix E.
>
> Thus, in our baseline setup, RTLDistil learns a mapping:
> [
> \text{(RTL, specs)} ;\rightarrow; (\text{AT}, \text{WNS}, \text{TNS})_{\text{post-layout}}.
> ]
>
> **What the dataset enables beyond this paper.**
> Crucially, CircuitNet 3.0 provides **richer modalities** than what RTLDistil currently consumes:
>
> * RTL text and SOG graphs,
> * gate-level netlists with structural features, and
> * layout-level information (e.g., density, placement, routing features).
>
> This allows future work to define **richer input configurations**, such as:
>
> * `(RTL + netlist)` → post-layout timing, or
> * `(RTL + rough floorplan / utilization)` → post-layout timing,
>
> under same **ground-truth labels** and tool flow. In this sense, CircuitNet 3.0 is a **general benchmark substrate** for early timing prediction with varying degrees of physical context.
>
> We will clarify this in the revised manuscript, encourage the kind of physically-informed timing prediction that the reviewer envisions.

---

### Author Response · Authors · 2025-11-26

We sincerely thank the Program Chairs, Area Chair, and all reviewers for their careful reading of our manuscript and for the thoughtful, constructive feedback. Your comments have been extremely valuable in helping us improve both the technical content and the presentation of this work.

We apologize for the slight delay in submitting this rebuttal. The additional time was spent running several new experiments (including ablations on data augmentation and multi-modal learning), performing large-scale consistency checks on the dataset, and updating the anonymous repository to better document our pipeline and artifacts. These efforts were aimed at addressing each concern as rigorously and concretely as possible.

In what follows, we respond to all comments point by point, grouped by reviewer. We very much appreciate this opportunity for dialogue, and we hope that the new results, clarifications, and added details will address your concerns and merit your further consideration. We'd sincerely hope to have your approval.

---

### Author Response · Authors · 2025-12-03
**Final Rebuttal Summary for CircuitNet 3.0**

We would like to express our sincere gratitude to the Program Chairs, Area Chairs, and all reviewers for their thoughtful and detailed reviews. Your feedback has been instrumental in shaping the final version of our work, and we greatly appreciate your time and effort in evaluating our submission.

We are pleased that our responses and additional experiments have addressed the concerns raised, and we will make every effort to incorporate the suggested improvements into the final manuscript. Specifically, we have:

1. Conducted experiments to validate the importance of task-oriented data augmentation versus mere data volume increase, demonstrating that our augmentation strategy significantly improves performance across both timing and power prediction tasks.
2. Addressed concerns related to multi-modal learning by adding controlled ablation studies for both timing and power prediction tasks, clearly showing the benefits of incorporating multi-modal information.
3. Clarified the dataset's purpose and scope, including the types of circuits it covers and its alignment with both early-stage EDA tasks and broader design objectives.
4. Provided detailed explanations of how our augmentation framework differs from prior work, emphasizing its focus on PPA-driven "adversarial" example generation tailored to hardware design.

We have also updated the repository with more comprehensive data and examples, ensuring better transparency and reproducibility for future research.

Our goal with **CircuitNet 3.0** is to create a comprehensive, task-oriented benchmark for machine learning-driven EDA that enables efficient, early-stage prediction of timing and power, and we believe the improvements and clarifications made during the rebuttal process further enhance the quality and impact of our work.

We hope that the revisions meet your expectations and that our contributions will be of value to the broader research community. We look forward to further feedback and the possibility of continuing this important dialogue.

Once again, thank you for your valuable input, and we appreciate your continued support and engagement with our work.

---

### Public Comment · ~Kyungjun_Min2 · 2026-02-25
**Suggestion to Include a Relevant Reference on AST-Based Verilog Augmentation**

Dear Authors,

I really enjoyed reading your paper and appreciate the tremendous effort put into creating CircuitNet 3.0. The multi-modal approach and the task-oriented augmentation framework are particularly impressive and valuable to the AI-for-EDA community.

While reviewing the section on the Verilog AST-based data augmentation framework, I noticed a significant technical overlap with our recently accepted paper at DATE 2025, titled "Improving LLM-Based Verilog Code Generation with Data Augmentation and RL." In our work, we also introduce an AST-based Verilog code augmentation technique using PyVerilog. Specifically, we leverage strategies such as node modification, insertion, and deletion to traverse and manipulate the tree structure for generating synthetic Verilog data.

Given the clear similarities in our approaches to Verilog AST manipulation for expanding ML datasets, I would be grateful if you could consider citing and briefly discussing our concurrent work in your camera-ready version. I believe acknowledging these concurrent efforts would provide readers with a more comprehensive background on the latest developments in AST-based augmentation techniques within the hardware design domain.

Thank you for your great contribution to the open-source EDA community, and I look forward to reading your final version.

Best regards,
Kyungjun Min

---

### Meta-Review · Area_Chair_H9CN · 2026-01-07

**Summary:**

The paper proposes CircuitNet 3.0, a comprehensive multi-stage and multi-modal dataset that enables advanced AI-driven circuit design through cross-stage data augmentation and filtering. All reviewers agree that the dataset is comprehensive and the empirical evaluations are thorough. Moreover, the paper is well written and easy to follow. The authors’ rebuttal adequately addressed most of the reviewers’ concerns, and the majority of reviews tend to accept the paper during the reviewer–author discussion phase. Therefore, I recommend this paper for acceptance at ICLR 2026. However, since the authors have not yet updated the manuscript, I require that the clarifications, refinements, and improvements discussed will be fully incorporated into the final camera-ready version.

**Reviewer Concerns:**

The reviewers raised substantial concerns during the review process. Although the authors adequately resolved most of them in their response, they did not provide a revised manuscript to reflect these improvements. Examples of necessary updates are as follows.
1. Experimental Specifications (raised by Reviewer cZe1). To ensure reproducibility, the authors must provide the missing technical details regarding the experimental setup. This includes explicit specifications of the PDK/tech nodes, voltage, frequency, and corner settings used in the evaluations.

2. Additional Ablation Experiments (raised by Reviewer myKG). The authors must include the supplementary experimental results discussed during the rebuttal to substantiate their claims. It is essential to present the ablation studies that isolate the sources of performance gains.

3. Related Work & Benchmarking (raised by Reviewer myKG). As specifically requested, it would be beneficial for the authors to discuss the relationship with recent benchmarks such as ChiPBench[2], and technically differentiate the proposed augmentation from existing software engineering method[1]. This comparison is crucial for positioning the contribution within the current landscape.

[1] Petrović, Goran, et al. "Practical mutation testing at scale: A view from google." IEEE Transactions on Software Engineering 48.10 (2021): 3900-3912.

[2] Wang, Zhihai, et al. Benchmarking end-to-end performance of ai-based chip placement algorithms. The Thirty-ninth Annual Conference on Neural Information Processing Systems Datasets and Benchmarks Track.

**Reviewer Scores:**

I think the final score of the reviewers are 4,6,6,8.

---

### Decision · Program_Chairs · 2026-01-26

Accept (Poster)